# The FlhA linker mediates flagellar protein export switching during flagellar assembly

Yumi Inoue[1,5,6], Miki Kinoshita [1,6], Mamoru Kida[2], Norihiro Takekawa[2], Keiichi Namba [1,3,4], Katsumi Imada [2✉] & Tohru Minamino [1✉]

The flagellar protein export apparatus switches substrate specificity from hook-type to filament-type upon hook assembly completion, thereby initiating filament assembly at the hook tip. The C-terminal cytoplasmic domain of FlhA (FlhA$_C$) serves as a docking platform for flagellar chaperones in complex with their cognate filament-type substrates. Interactions of the flexible linker of FlhA (FlhA$_L$) with its nearest FlhA$_C$ subunit in the FlhA$_C$ ring is required for the substrate specificity switching. To address how FlhA$_L$ brings the order to flagellar assembly, we analyzed the *flhA(E351A/W354A/D356A) ΔflgM* mutant and found that this triple mutation in FlhA$_L$ increased the secretion level of hook protein by 5-fold, thereby increasing hook length. The crystal structure of FlhA$_C$(E351A/D356A) showed that FlhA$_L$ bound to the chaperone-binding site of its neighboring subunit. We propose that the interaction of FlhA$_L$ with the chaperon-binding site of FlhA$_C$ suppresses filament-type protein export and facilitates hook-type protein export during hook assembly.

[1] Graduate School of Frontier Biosciences, Osaka University, Suita, Osaka, Japan. [2] Department of Macromolecular Science, Graduate School of Science, Osaka University, Toyonaka, Osaka, Japan. [3] RIKEN SPring-8 Center and Center for Biosystems Dynamics Research, Suita, Osaka, Japan. [4] JEOL YOKOGUSHI Research Alliance Laboratories, Osaka University, Suita, Osaka, Japan. [5] Present address: Department of Ophthalmology and Visual Sciences, Kyoto University Graduate School of Medicine, Kyoto, Japan. [6] These authors contributed equally: Yumi Inoue, Miki Kinoshita. ✉email: kimada@chem.sci.osaka-u.ac.jp; tohru@fbs.osaka-u.ac.jp

The flagellum of *Salmonella enterica* (hereafter referred to as *Salmonella*) is a supramolecular motility machine consisting of the basal body, which acts as a bi-directional rotary motor, the hook, which functions as a universal joint, and the filament, which works as a helical propeller[1]. For construction of the flagella on the cell surface, the flagellar type III secretion system (fT3SS) transports flagellar building blocks from the cytoplasm to the distal end of the growing flagellar structure[2]. The fT3SS is divided into three structural parts: a transmembrane export gate complex made of FlhA, FlhB, FliP, FliQ, and FliR, a substrate-chaperone-docking platform composed of the cytoplasmic domains of FlhA and FlhB (FlhA$_C$ and FlhB$_C$), and a cytoplasmic ATPase ring complex consisting of FliH, FliI, and FliJ[3]. The FlhA$_C$–FlhB$_C$-docking platform provides binding sites for the cytoplasmic ATPase complex, flagellar export chaperones (FlgN, FliS, FliT), and export substrates to mediate hierarchical protein targeting and secretion[4].

Flagellar assembly begins with the basal body, followed by the hook (FlgE) with the help of the hook cap (FlgD). After completion of hook–basal body (HBB) assembly, the FlgD cap is replaced by FlgK, and then FlgK and FlgL form the hook–filament junction structure at the hook tip, followed by the assembly of the filament cap (FliD). Finally, newly transported flagellin molecules (FliC) assemble into the filament with the help of the filament cap (Fig. 1)[5]. Flagellar building blocks are classified into two export classes: one is the rod-type (FliE, FlgB, FlgC, FlgF, FlgG, FlgJ) and hook-type class (FlgD, FlgE, and FliK) needed for the assembly of the rod and hook, and the other is the filament-type class (FlgK, FlgL, FlgM, FliC, and FliD) responsible for filament assembly at the hook tip[6,7]. The FlhA$_C$–FlhB$_C$-docking platform serves as an export switch to induce substrate specificity switching from rod-/hook-type proteins to filament-type ones when the hook reaches its mature length of about 55 nm in *Salmonella*, thereby terminating hook assembly and initiating filament formation (Fig. 1)[8–11].

The fT3SS uses a secreted molecular ruler protein (FliK) to measure the hook length during hook assembly[4]. FliK is a hook-type protein secreted via the fT3SS during HBB assembly[12]. FliK not only measures the hook length[13–15] but also switches substrate specificity of the FlhA$_C$–FlhB$_C$-docking platform (Fig. 1)[11,16,17]. This has been recently verified by in vitro reconstitution experiments using inverted membrane vesicles[18,19]. The N-terminal domain of FliK (FliK$_N$) acts as a secreted molecular ruler to measure the hook length[13–15]. When the hook length reaches about 55 nm, a flexible linker region of FliK connecting FliK$_N$ and the C-terminal domain (FliK$_C$) promotes a conformational rearrangement of FliK$_C$, allowing FliK$_C$ to interact with FlhB$_C$ to terminate the export of the rod-type and hook-type proteins[20,21].

FlhA$_C$ (residues 328–692) consists of four domains, D1, D2, D3, and D4, and a flexible linker (FlhA$_L$) (residues 328–361) connecting FlhA$_C$ with the N-terminal transmembrane domain of FlhA (Fig. 2a)[22]. FlhA$_C$ forms a homo-nonamer ring in the fT3SS[23] and provides binding sites for flagellar export chaperons (FlgN, FliS, and FliT) in complex with their cognate filament-type proteins (Fig. 2a)[24–27]. The flagellar chaperones promote the docking of their cognate filament-type substrates to the FlhA$_C$ ring structure to facilitate subsequent unfolding and translocation of the substrates[28,29]. High-speed atomic force microscopy combined with mutational analysis has shown that FlhA$_L$ is required for highly cooperative FlhA$_C$ ring formation on mica surface[10]. Glu-351, Trp-354, and Asp-356 of FlhA$_L$ bind to the D1 and D3 domains of its neighboring FlhA$_C$ subunit to stabilize FlhA$_C$ ring structure (Fig. 2a)[10], and the W354A, E351A/D356A, and E351A/W354A/D356A mutations in FlhA$_L$ not only inhibit FlhA$_C$ ring formation but also reduce the binding affinity of

FlhA$_C$ for flagellar chaperones in complex with their cognate filament-type substrates, thereby inhibiting the initiation of filament assembly[10]. Therefore, the FliK$_C$–FlhB$_C$ interaction is postulated to modify the binding mode of FlhA$_L$ to its nearest subunit in the FlhA$_C$ ring structure upon completion of the hook structure, thereby allowing the flagellar chaperones to bind to FlhA$_C$ to initiate the export of filament-type proteins[10,11,30]. However, it remains unknown how FlhA$_L$ regulates the interactions of FlhA$_C$ with the chaperones during HBB assembly.

In the present study, to clarify the role of FlhA$_L$ in the export switching mechanism of fT3SS, we analyzed the interaction between FlhA$_L$ and FlhA$_C$ and provide evidence suggesting that the interaction of FlhA$_L$ with the chaperone-binding site of FlhA$_C$ brings the order to flagellar protein export in parallel with the assembly order of the flagellar structure.

## Results

**Isolation of pseudorevertants from the *flhA(E351A/W354A/D356A)* mutant.** Glu-351, Trp-354, and Asp-356 of FlhA$_L$ bind to the D1 and D3 domains of its neighboring FlhA$_C$ subunit to stabilize FlhA$_C$ ring structure (Fig. 2a)[10]. The *flhA(E351A/D356A)* (hereafter referred to as *flhA$_{ED}$*) and *flhA(W354A)* (hereafter referred to as *flhA$_W$*) mutants produce the HBBs without the filament attached[10]. Hook lengths of the *flhA$_{ED}$* and *flhA$_W$* mutants are 54.0 ± 22.3 nm [mean ± standard deviation (SD)] and 52.9 ± 19.9 nm, respectively, where their SD values are larger than the wild-type one (51.0 ± 6.9 nm), indicating their hook length is not controlled properly[10]. Pull-down assays by GST affinity chromatography have revealed that the *flhA$_{ED}$* and *flhA$_W$* mutations reduce the binding affinity of FlhA$_C$ for flagellar chaperones in complex with their cognate filament-type substrates[10]. These previous results suggest that the observed interaction between FlhA$_L$ and the D1 and D3 domains of its neighboring FlhA$_C$ subunit is responsible for making the chaperone-binding site of FlhA$_C$ open to allow the chaperones to bind to FlhA$_C$ to facilitate the export of filament-type proteins[10]. However, the *flhA(E351A/W354A/D356A)* (hereafter referred to as *flhA$_{EWD}$*) mutant does not produce the HBBs[10], and this raises a question as to why the *flhA$_{EWD}$* mutation inhibits HBB assembly.

To address this question, we first carried out quantitative immunoblotting to measure the amount of flagellar building blocks secreted by the fT3SS. The *flhA$_{EWD}$* mutation significantly reduced the secretion levels of both hook-type (FlgD, FlgE, FliK) and filament-type substrates (FlgK, FliC, FliD) (Fig. 2b, c), indicating that the *flhA$_{EWD}$* mutation significantly reduces the protein transport activity of the fT3SS.

To clarify why and how the *flhA$_{EWD}$* mutation inhibits flagellar protein export, we isolated 14 pseudorevertants from the *flhA$_{EWD}$* mutant. Motility of the pseudorevertants was somewhat better than that of the *flhA$_{EWD}$* mutant but was much poorer than that of wild-type cells (Supplementary Fig. 1a). Export substrates such as FlgD, FlgE, FlgK, and FliD were detected in the culture supernatants of these pseudorevertants (Supplementary Fig. 1b). Consistently, these pseudorevertants produced a couple of flagella on the cell surface (Supplementary Fig. 1c). DNA sequencing revealed that all suppressor mutations are located in the *flgMN* operon. One was the M1I mutation at the start codon of the *flgM* gene (isolated twice), presumably inhibiting FlgM translation. Two suppressor mutations produced a stop codon at position of Gln-52 or Ser-85 of FlgM, resulting in truncation of the C-terminal region of FlgM. Nine suppressor mutations were large deletions in *flgM*. We also found that there was a large deletion in the *flgM* and *flgN* genes, thereby disrupting both FlgM and FlgN. A loss-of-function of FlgM results in a considerable increment in the transcription levels of flagellar genes[31]. Consistently, the

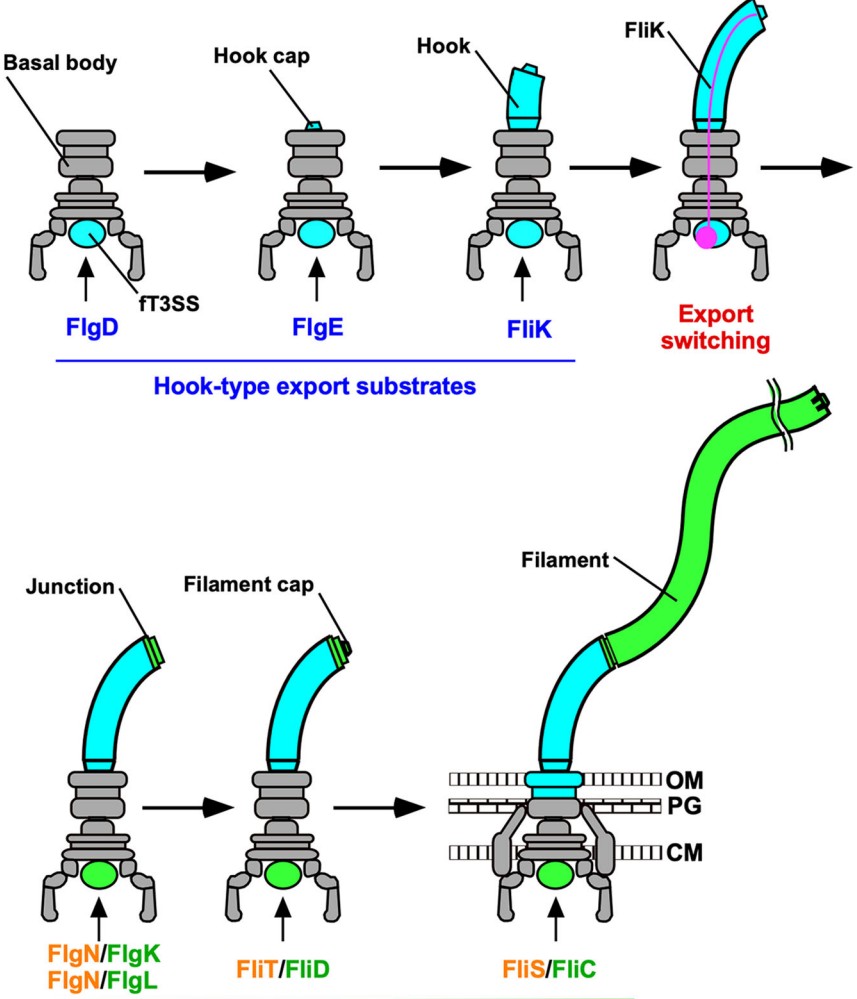

**Fig. 1 Flagellar assembly pathway.** The *Salmonella* flagellum is composed of the basal body, the hook, the hook–filament junction, the filament and the filament cap. Upon completion of basal body assembly, newly exported FlgE molecules polymerize into the hook structure with the help of the hook cap made of FlgD. When the hook reaches its mature length of about 55 nm, the hook cap is replaced by FlgK. FlgK and FlgL self-assemble at the hook tip in this order to form the junction structure. Then, FliD forms the filament cap at the tip of the junction and promotes the assembly of FliC into the filament. A type III protein export apparatus (fT3SS) is located at the flagellar base and transports flagellar building blocks from the cytoplasm to the distal end of the growing flagellar structure. The fT3SS sometimes secretes the FliK ruler to measure the hook length during hook assembly. When the hook reaches its mature length of about 55 nm, the fT3SS switches its substrate specificity, thereby terminating the export of hook-type proteins (FlgD, FlgE, and FliK) and initiating the export of filament-type proteins (FlgK, FlgL, FliD, and FliC). FlgN, FliT, and FliS act as flagellar type III export chaperones specific for FlgK and FlgL, FliD and FliC, respectively. OM outer membrane, PG peptidoglycan layer, CM cytoplasmic membrane.

cellular levels of flagellar building blocks and the FliI ATPase were higher in the pseudorevertants than those in its parental strain (Fig. 2b).

The interaction of FliJ with FlhA$_L$ is required for activation of the fT3SS, and FliH and FliI are required for efficient interaction between FliJ and FlhA$_L$[32]. It has been reported that over-expression of export substrates and FliJ by FlgM deletion overcomes the loss of both FliH and FliI to a considerable degree[33]. Because the *flhA$_{EWD}$* mutation reduces the binding affinity of FlhA$_C$ for FliJ[10], this suggests that these *flgM* mutations increase the cytoplasmic levels of FliH, FliI, FliJ, and export substrates to allow the *flhA$_{EWD}$* mutant to export flagellar building blocks for producing a small number of flagella on the cell surface. Therefore, we propose that Glu-351, Trp-354, and Asp-356 of FlhA$_L$ also play an important role in the activation mechanism of the fT3SS.

**Effect of deletion of FlhA$_L$ on the interaction between FlhA$_C$ and FliJ.** The crystal structure of a FliJ homolog, CdsO, in complex with CdsV$_C$, which is a FlhA$_C$ homolog, has shown that CdsO binds to a large cleft between domains D4 of neighboring CdsV$_C$ subunits in the CdsV$_C$ ring structure but not to the linker region of CdsV$_C$[34] (Fig. 2a). To confirm the importance of FlhA$_L$ in the interaction between FlhA$_C$ and FliJ, we analyzed the binding of FlhA$_C$ to immobilized GST-FliJ by Bio-layer inter-ferometry (BLI) measurements[35]. The FliJ–FlhA$_C$ interaction showed a complex binding profile (Fig. 3, 1st row) and did not fit the global one-state association-then-dissociation model. Assuming that FlhA$_C$ binds to GST-FliJ to form a GST-FliJ/FlhA$_C$ complex, followed by a conformational change of this complex, the BLI data fitted well with a two-state reaction model and provided a $K_D$ value of $1.36 \pm 0.03$ μM (mean ± SD, $n = 3$). Unlike wild-type FlhA$_C$, the association and dissociation processes of

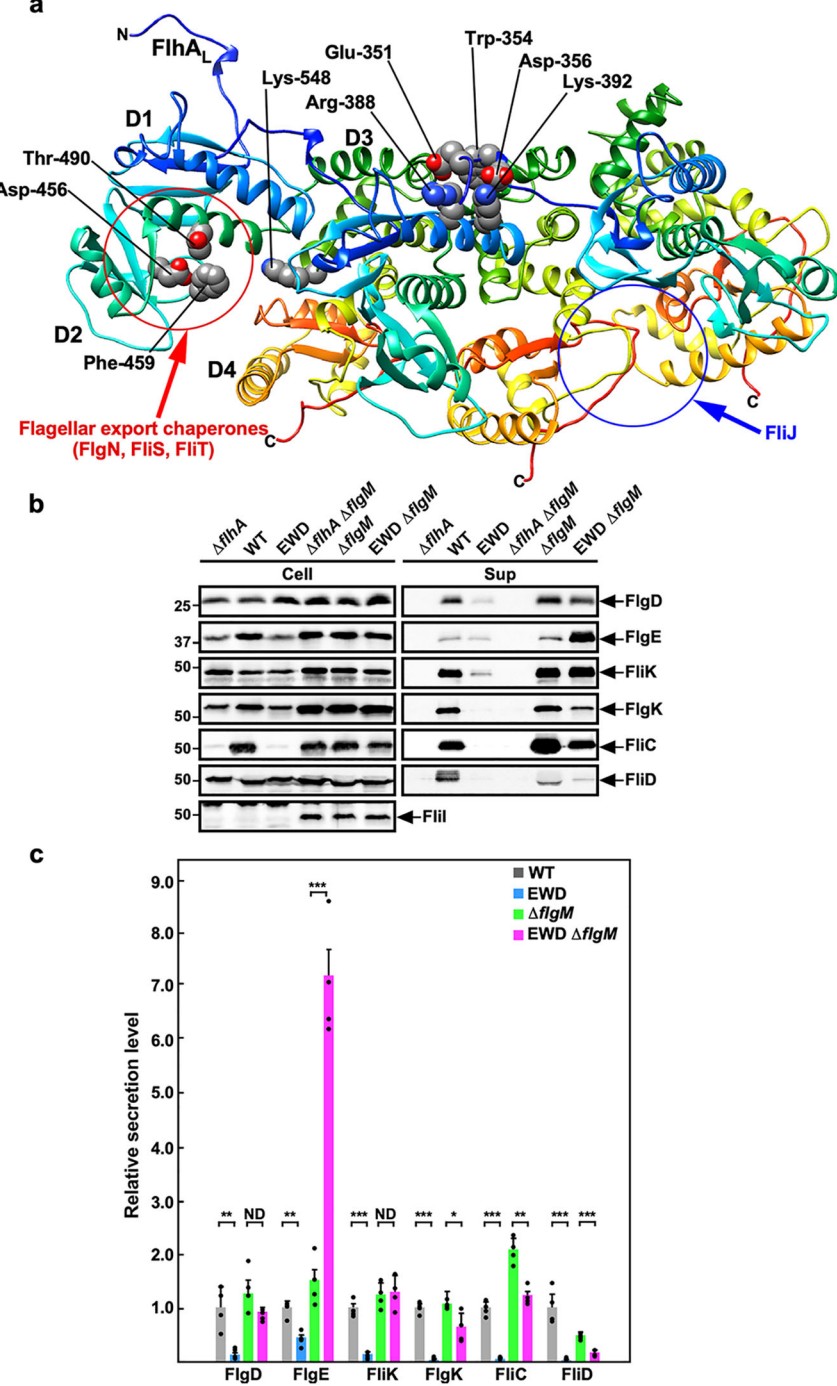

**Fig. 2 Effect of the *flhA_EWD* mutation on flagellar protein export. a** Structural model of the FlhA_C ring. Only three FlhA_C subunits in the FlhA_C nonameric ring model are shown. FlhA_C (PDB ID: 3A5I) consists of four domains, D1, D2, D3, and D4 and a flexible linker (FlhA_L). Glu-351, Trp-354, and Asp-356 of FlhA_L binds to the D1 and D3 domains of its neighboring subunit. A well-conserved hydrophobic dimple including Asp-456, Phe-459, and Thr-490 is responsible for the interaction of FlhA_C with flagellar export chaperones in complex with filament-type substrates. Phe-459 and Lys-548 are exposed to solvent on the molecular surface when FlhA_C adopts the open conformation. FliJ binds not only to FlhA_L but also to a large cleft between the D4 domains. **b** Immunoblotting, using polyclonal anti-FlgD (1st row), anti-FlgE (2nd row), anti-FliK (3rd row), anti-FlgK (4th row), anti-FliC (5th row), anti-FliD (6th row), or anti-FliI (7th row) antibody of whole-cell proteins and culture supernatant fractions prepared from the *Salmonella* NH001 strain transformed with pTrc99AFF4 (Δ*flhA*), pMM130 (WT), or pYI003 (EWD) and the NH001gM strain transformed with pTrc99FF4A (Δ*flhA* Δ*flgM*), pMM130 (Δ*flgM*), or pYI003 (EWD Δ*flgM*). The positions of molecular mass markers are indicated on the left. The regions of interest were cropped from original immunoblots shown in Supplementary Fig. 7. **c** Relative secretion levels of flagellar proteins. These data are the average of four independent experiments. The average density of each flagellar protein seen in the culture supernatant derived from wild-type cells was set to 1.0, and then relative band density was calculated. Vertical bars indicate standard deviations. Dots indicate individual data points. The source data are shown in Supplementary Data File. Comparisons between datasets were performed using a two-tailed Student's *t*-test. A *P* value of <0.05 was considered to be statistically significant difference. *$P$ < 0.05; *$P$ < 0.01; **$P$ < 0.001; ND no statistical difference.

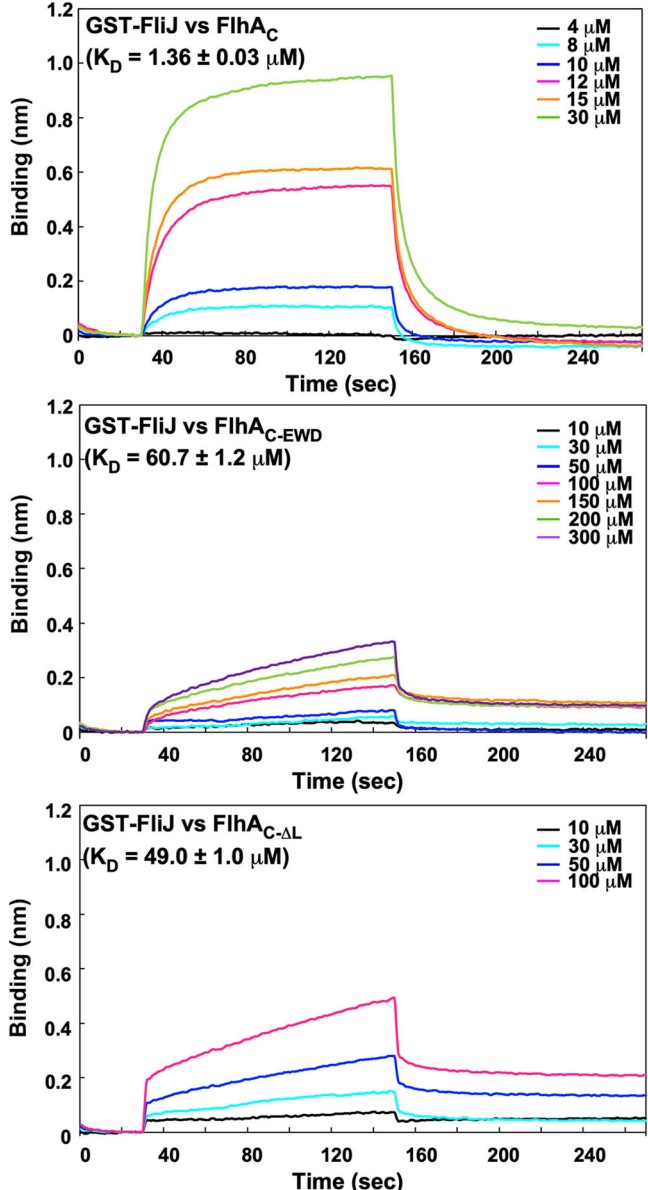

**Fig. 3 Effect of FlhA linker mutations on the interaction of FlhA_C with FliJ.** BLI profiles were obtained from the FlhA_C–FliJ interaction (1st row), the FlhA_C-EWD–FliJ interaction (2nd row), and the FlhA_C-ΔL–FliJ interaction (3rd row). GST-FliJ was immobilized to an anti-GST sensor tip. The sensor tip was then dipped into FlhA_C, FlhA_C-EWD, or FlhA_C-ΔL of various concentrations to measure association before being dipped into the kinetic buffer to measure dissociation. Three independent measurements were carried out. All experiments were performed at 25 °C.

FlhA_C with the *flhA_EWD* mutation (FlhA_C-EWD) or FlhA_C lacking FlhA_L (FlhA_C-ΔL) were observed only at protein concentrations above 10 μM (Fig. 3, 2nd and 3rd rows). Their association and dissociation processes were also different from those of wild-type FlhA_C. The association profile of these mutant proteins was composed of two distinct (fast-on and slow-on) processes, and the dissociation profile was also composed of two distinct (fast-off and slow-off) processes. It has been shown that wild-type FlhA_C forms dimer in a protein concentration-dependent manner and that FlhA_L is required for efficient dimerization of FlhA_C[27]. So, their BLI data were fitted well with curves predicted by the Hill equation, with $K_D$ values of 60.7 ± 1.2 μM ($n = 3$) and 49.0 ± 1.0 μM ($n = 3$) for the FliJ–FlhA_C-EWD and FliJ–FlhA_C-ΔL

interactions, respectively. Thus, both *flhA_EWD* mutation and deletion of FlhA_L reduced the binding affinity of FlhA_C for FliJ. Therefore, we conclude that FlhA_L is required for the stable interaction between FliJ and FlhA_C.

**Effect of the *flhA_EWD* mutation on flagellar protein export by fT3SS in the absence of FlgM.** To quantify the amount of flagellar building blocks secreted by the *flhA_EWD* Δ*flgM* strain, we introduced the Δ*flgM::km* allele to the *Salmonella* NH001 (Δ*flhA*) strain to produce the Δ*flgM* and *flhA_EWD* Δ*flgM* cells. The Δ*flgM:: km* allele restored motility of the *flhA_EWD* mutant in a way similar to other *flgM* suppressor mutations (Supplementary Fig. 2a, b). The amount of FlgE secreted by the *flhA_EWD* Δ*flgM* strain was about fivefold higher than that by the Δ*flgM* strain (Fig. 2b, c), suggesting that this triple mutation significantly increases the binding affinity of the fT3SS for FlgE. However, the *flhA_EWD* mutation did not affect the levels of FlgD and FliK secretion (Fig. 2b, c). These observations suggest that FlhA_L may regulate substrate recognition of the fT3SS for hierarchical protein targeting and secretion among the hook-type substrates. The amount of FlgK, FliC, and FliD secreted by the *flhA_EWD* Δ*flgM* strain was significantly lower than that by the Δ*flgM* strain (Fig. 2b, c), indicating that the *flhA_EWD* mutation also affects export switching of the fT3SS from hook-type substrates to filament-type ones.

We found that the *flhA_EWD* Δ*flgM* strain secreted a much larger amount of FlgE into the culture media than the Δ*flgM* strain, raising the possibility that the length of the hook produced by this mutant may be longer than the wild-type length. To clarify this, we isolated flagella from the Δ*flgM* and *flhA_EWD* Δ*flgM* cells and measured their hook length. The hook length of the Δ*flgM* strain was 52.0 ± 5.1 nm (mean ± SD, $n = 157$) (Fig. 4, left panels), which is nearly the same as that of the wild-type strain (51.0 ± 6.9 nm)[10]. This indicates that the loss-of-function mutation of FlgM does not affect the hook length control. In contrast, the average hook length of the *flhA_EWD* Δ*flgM* strain was 68.8 ± 30.9 nm ($n = 122$) (Fig. 4, right panels), indicating that the hook length control becomes worse in the presence of the *flhA_EWD* mutation. These suggest that this mutation affects not only the initiation of filament-type protein export but also the termination of hook-type protein export. Because high-speed atomic force microscopy has shown that the *flhA_EWD* mutation also inhibits highly cooperative FlhA_C ring formation[10], we propose that FlhA_L regulates the conformational rearrangement of FlhA_C in the ring, which is required for efficient termination of hook assembly and efficient initiation of filament formation at the hook tip.

**Effect of FlhA linker mutations on the hydrodynamic properties of FlhA_C in solution.** A well-conserved hydrophobic dimple of FlhA_C containing Asp-456, Phe-459, and Thr-490 residues is located at the interface between domains D1 and D2 and is involved in the interactions with the FlgN, FliS, and FliT chaperones in complex with their cognate filament-type substrates (Fig. 2a)[25–27]. The *flhA_W*, *flhA_ED*, and *flhA_EWD* mutations reduce the binding affinity of FlhA_C for these chaperone/substrate complexes[10]. Interestingly, the *flhA(D456V)*, *flhA(F459A)*, and *flhA(T490M)* mutations increase the secretion levels of FlgE and FliK by the Δ*fliH-fliI flhB(P28T)* mutant[36]. We found that the *flhA_EWD* mutation increases the secretion level of FlgE by about fivefold, raising the possibility that FlhA_L carrying either of *flhA* linker mutations binds to the hydrophobic dimple of FlhA_C not only to facilitate the export of FlgE but also to block the FlhA_C–chaperone interaction. If this is the case, FlhA_C with these mutations would show distinct hydrodynamic properties compared with wild-type FlhA_C. To clarify this possibility, we

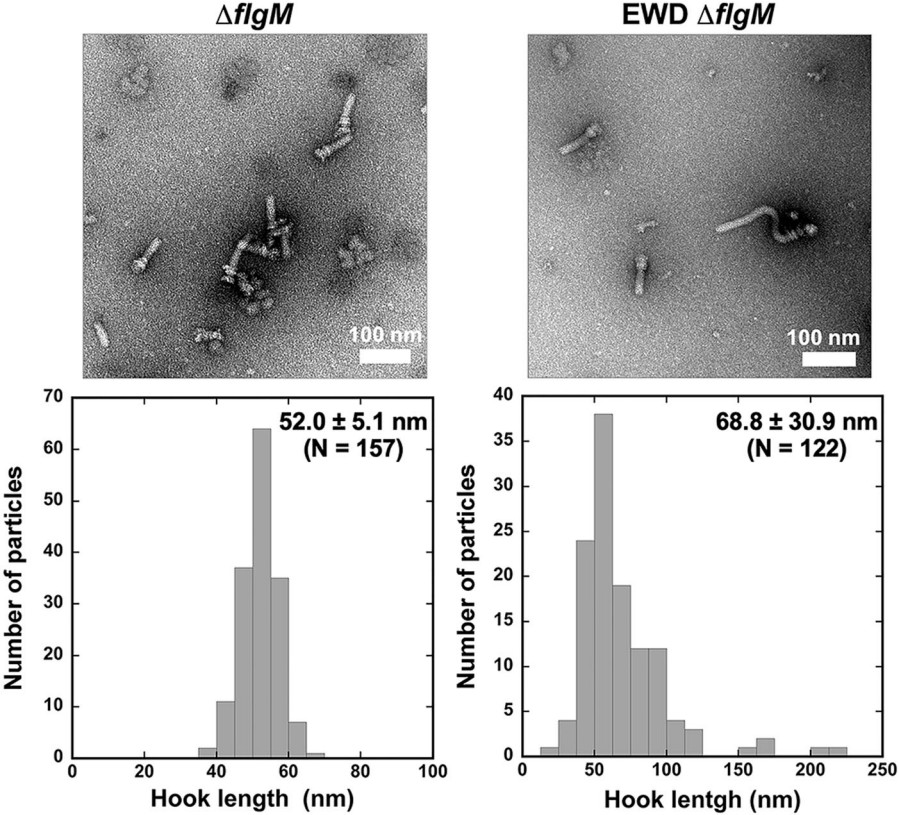

**Fig. 4 Effect of the *flhA_EWD* mutation on hook length.** Electron micrographs of HBBs and histograms of hook length distribution of NH001gM carrying pMM130 (Δ*flgM*) or pYI003 (EWD Δ*flgM*).

performed size exclusion chromatography (SEC) with a Superdex 75 column HR 10/30 column. Wild-type His-FlhA$_C$ appeared as a single peak at an elution volume of 10.2 ml, which corresponds to the deduced molecular mass of His-FlhA$_C$ (about 43 kDa) (Fig. 5a). His-FlhA$_C$ with the *flhA_W* (FlhA$_{C-W}$), *flhA_ED* (FlhA$_{C-ED}$) or *flhA_EWD* mutation (FlhA$_{C-EWD}$) and FlhA$_{C-ΔL}$ lacking FlhA$_L$ appeared as a single peak at an elution volume of 10.3, 10.5, 10.4, and 11.0 ml, respectively (Fig. 5a), indicating that these mutant variants exist as a monomer in solution. FlhA$_{C-ED}$ exhibited a delayed elution behavior compared with the wild type. Furthermore, FlhA$_{C-ED}$ showed a slightly faster mobility in both sodium dodecyl sulfate polyacrylamide gel electrophoresis (SDS-PAGE) and native PAGE gels (Fig. 5b, c). Far-UV CD measurements revealed that the *flhA_ED* mutation did not affect the secondary structures of FlhA$_C$ (Supplementary Fig. 3). These suggest that FlhA$_{C-ED}$ adopts a more compact conformation than wild-type FlhA$_C$. The elution peak position of FlhA$_{C-EWD}$ was between those of the wild type and FlhA$_{C-ED}$ (Fig. 5a). Because FlhA$_{C-EWD}$ showed two different bands on SDS-PAGE gels, with a slower mobility band corresponding to wild-type FlhA$_C$ and a faster one corresponding to FlhA$_{C-ED}$ (Fig. 5b), we suggest that FlhA$_{C-EWD}$ exists in an equilibrium between the wild-type conformation and the compact conformation. Therefore, we suggest that the *flhA_ED* mutation is required to make FlhA$_C$ more compact.

**Effect of FlhA linker mutations on methoxypolyethylene glycol 5000 maleimide (mPEG-maleimide) modifications of Cys-459 and Cys-548.** FlhA$_C$ structures adopt open, semi-closed, and closed

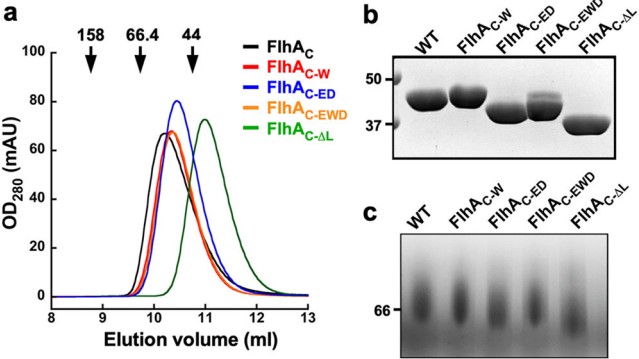

**Fig. 5 Effect of FlhA linker mutations on the FlhA$_C$ conformation. a** Hydrodynamic properties of FlhA$_C$ and its FlhA linker mutant variants. Purified protein samples (10 μM) were run on a Superdex 75HR 10/30 column equilibrated with 50 mM Tri-HCl, pH 8.0, 150 mM NaCl. The elution peaks of His-FlhA$_C$ (WT, black), His-FlhA$_{C-W}$ (red), His-FlhA$_{C-ED}$ (blue), His-FlhA$_{C-EWD}$ (orange), and His-FlhA$_{C-ΔL}$ (green) are 10.2, 10.3, 10.5, 10.4, and 11.0 ml, respectively. Arrow indicates the elution peaks of γ-globulin (158 kDa), bovine serum albumin (66.4 kDa), and ovalbumin (43 kDa), which are 8.7, 9.7, and 10.7 ml, respectively. **b** CBB-stained SDS-PAGE gel of purified wild-type FlhA$_C$ and its mutant variants. The regions of interest were cropped from an original CBB-stained gel shown in Supplementary Fig. 8a. **c** Blue Native PAGE gel of purified wild-type FlhA$_C$ and its mutant variants. The regions of interest were cropped from an original Blue Native PAGE gel shown in Supplementary Fig. 8b.

**Table 1 Data collection and refinement statistics.**

| | FlhA$_C$(E351A/D356A) |
|---|---|
| **Data collection** | |
| Space group | $P2_12_12_1$ |
| Cell dimensions | |
| $a, b, c$ (Å) | 71.7, 96.2, 114.1 |
| $\alpha, \beta, \gamma$ (°) | 90.0, 90.0, 90.0 |
| Resolution (Å) | 73.5–2.80 (2.95–2.80)[a] |
| $R_{merge}$ | 0.074 (0.317) |
| CC(1/2) | 0.995 (0.915) |
| $I / \sigma I$ | 8.1 (2.8) |
| Completeness (%) | 97.1 (94.6) |
| Redundancy | 3.4 (3.1) |
| **Refinement** | |
| Resolution (Å) | 73.5–2.80 (2.87–2.80) |
| No. of reflections | 19,301 (1294) |
| $R_{work}/R_{free}$ | 23.2/29.0 (33.5/42.1) |
| No. of atoms | |
| Protein | 5252 |
| Ligand/ion | 0 |
| Water | 0 |
| B-factors | |
| Protein | 70.0 |
| Ligand/ion | – |
| Water | – |
| R.m.s. deviations | |
| Bond lengths (Å) | 0.003 |
| Bond angles (°) | 0.680 |

Number of crystals: 1.
[a]Values in parentheses are for highest-resolution shell.

conformations[22–24,27,30,37]. A large open cleft between domains D2 and D4 is seen in the open form, but not in the closed form. As a result, Phe-459 and Lys-548, which are both located in the cleft between domains D2 and D4, are fully exposed to solvent on the molecular surface of the open conformation of FlhA$_C$ but are in close proximity to each other in the closed conformation[22,30,37]. To test whether mutations in FlhA$_L$ bias FlhA$_C$ towards the closed structure, we performed Cys modification experiments with mPEG-maleimide. FlhA$_C$ with the F459C/K548C substitutions modified by mPEG-maleimide showed much slower mobility shift (Supplementary Fig. 4, left panel), in agreement with a previous report[30]. The *flhA$_W$*, *flhA$_{ED}$*, and *flhA$_{EWD}$* mutations did not inhibit Cys modifications with mPEG-maleimide (Supplementary Fig. 4, right panel), indicating that FlhA$_C$ with these mutations does not adopt the closed conformation.

**Crystal structure of FlhA$_{C-ED}$.** To investigate whether FlhA$_L$ binds to the hydrophobic dimple of FlhA$_C$ to make FlhA$_{C-ED}$ more compact, we explored crystallization conditions of FlhA$_{C-ED}$ for a molecular packing distinct from the open (PDB code: 3A5I)[22] and semi-closed (PDB code: 6AI0)[30] forms of wild-type FlhA$_C$. We found a new orthorhombic crystal that diffracted up to 2.8 Å resolution, with unit cell dimensions $a = 71.7$ Å, $b = 96.2$ Å, $c = 114.1$ Å (Table 1) and the asymmetric unit containing two FlhA$_C$ molecules (A and B). Mol-A adopts an open conformation similar to the 3A5I structure (Supplementary Fig. 5a, c) whereas Mol-B shows a semi-closed conformation similar to the 6AI0 structure (Supplementary Fig. 5b, d). The residues from Val-349 to Val-357 in FlhA$_L$ of Mol-A form an α-helix, which interacts with the hydrophobic dimple of a neighboring Mol-A molecule related by a crystallographic symmetry (Fig. 6a).

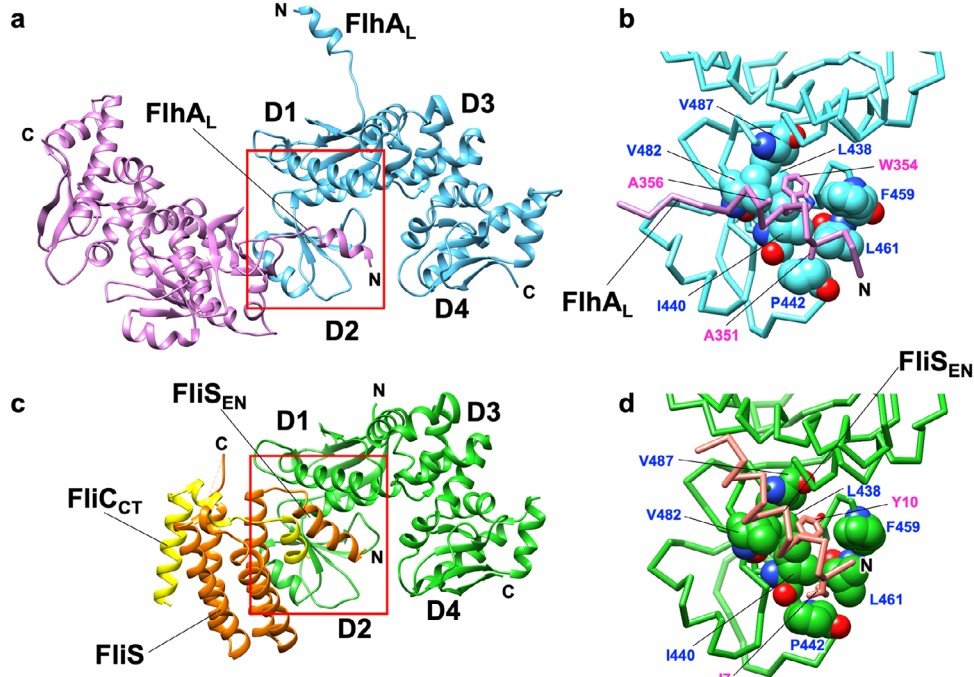

**Fig. 6 Interaction between FlhA$_L$ and a well-conserved hydrophobic dimple of its neighboring FlhA$_C$ in the crystal of FlhA$_{C-ED}$. a** FlhA$_L$ of Mol-A (magenta) interacts with neighboring Mol-A (cyan) related by a crystallographic symmetry. **b** Close-up view of the interaction between FlhA$_L$ and the hydrophobic dimple shown by a red box in **a**. Residues that form the hydrophobic dimple are indicated by balls. The side chains of Ala-351, Trp-354, and Ala-356 in FlhA$_L$ are shown in stick models. **c** Interaction between FlhA$_C$ (green) and FliS (orange) fused with the C-terminal region of FliC (yellow) (PDB code: 6CH3). **d** Close-up view of the interaction between the extreme N-terminal region of FliS (FliS$_{EN}$) and the hydrophobic dimple shown by a red box in **c**. The residues that form the hydrophobic dimple are indicated by ball. The side chains of Ile-7 and Tyr-10 of FliS$_{EN}$ are shown in stick models.

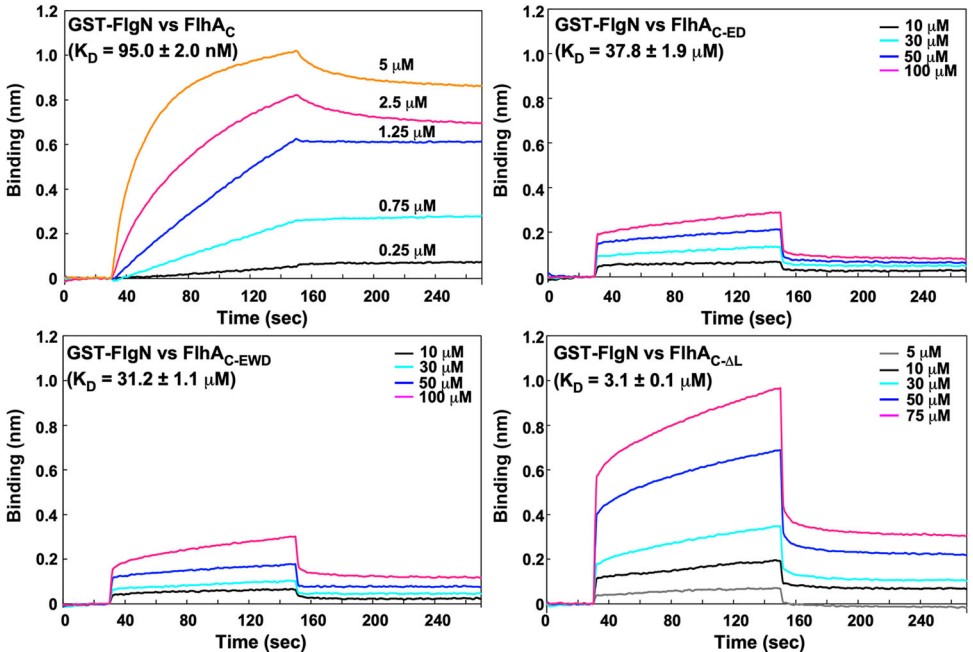

**Fig. 7 Effect of FlhA linker mutations on the interaction of FlhA$_C$ with FlgN.** BLI profiles were obtained from the FlhA$_C$–FlgN interaction (upper, left panel), the FlhA$_{C-ED}$–FlgN interaction (upper, right panel), the FlhA$_{C-EWD}$–FlgN interaction (lower, left panel) and the FlhA$_{C-ΔL}$–FlgN interaction (lower, right panel). GST-FlgN was immobilized to an anti-GST sensor tip. The sensor tip was then dipped into FlhA$_C$, FlhA$_{C-ED}$, FlhA$_{C-EWD}$, or FlhA$_{C-ΔL}$ of various concentrations to measure association before being dipped into the kinetic buffer to measure dissociation. Three independent measurements were carried out. All experiments were performed at 25 °C.

Trp-354 fits into the hydrophobic dimple, and Ala-351 hydrophobically contacts with Pro-442 on the periphery of the dimple (Fig. 6b and Supplementary Fig. 6a). These interactions resemble the interaction between the N-terminal α-helix of FliS and the hydrophobic dimple of FlhA$_C$ (PDB ID: 6CH3)[27] (Fig. 6c). Ile-7 and Tyr-10 of the N-terminal α-helix of FliS is in the corresponding position of Ala-351 and Trp-354 of FlhA$_L$, respectively. Tyr-10 fits into the hydrophobic dimple of FlhA$_C$, and Ile-7 interacts with Pro-442 of FlhA$_C$ (Fig. 6d). These observations suggest that FlhA$_L$ and flagellar chaperones bind competitively to a common binding site on FlhA$_C$ and that the dissociation of FlhA$_L$ from this binding site is required for the binding of the flagellar chaperones to FlhA$_C$. When Ala-351 and Ala-356 of FlhA$_{C-ED}$ in the crystal structure were replaced back to the original Glu-351 and Asp-356 residues, respectively, the side chain arm of Glu-351 can form a hydrophobic contact with Pro-442 (Supplementary Fig. 6b), suggesting that FlhA$_L$ can bind to the hydrophobic dimple of FlhA$_C$ even in the wild type without the $flhA_{ED}$ mutation. Because the introduced Ala residues would increase the helical propensity of residues 349–357 of FlhA$_L$ as seen in the crystal, we suggest that the $flhA_{ED}$ mutation allowed residues 349–357 of FlhA$_L$ to efficiently form an α-helix to stabilize the binding of FlhA$_L$ to the hydrophobic dimple of FlhA$_C$.

**Effect of FlhA linker mutations on the interaction of FlhA$_C$ with the FlgN chaperone.** We found that FlhA$_L$ with the $flhA_{ED}$ mutation binds to the chaperone-binding site in its neighboring subunit in the crystal. If this interaction reflects the functional state of FlhA$_C$, the $flhA_{ED}$ mutation would affect the docking process of FlgN to FlhA$_C$. To clarify this hypothesis, we performed BLI measurements. When GST-FlgN was tethered to a sensor chip and then allowed FlhA$_C$ of various concentrations to bind to immobilized GST-FlgN, the interaction between FlgN and FlhA$_C$ showed a typical BLI profile (Fig. 7). The association and

dissociation rate constants were measured to be about $8.23 ± 0.20 × 10^3 \, M^{-1} \, S^{-1}$ and $7.81 ± 0.03 × 10^{-4} \, S^{-1}$, respectively, giving a $K_D$ value of $95.0 ± 2.0 \, nM$ (mean ± SD, $n = 3$). This $K_D$ value is in agreement with previous data obtained by surface plasmon resonance[24]. Unlike wild-type FlhA$_C$, FlhA$_{C-ED}$ and FlhA$_{C-EWD}$ did not bind to immobilized GST-FlgN at protein concentrations less than 10 μM, indicating that FlhA$_L$ with either of these two mutations inhibits the binding of FlhA$_C$ to FlgN. The association and dissociation processes of FlhA$_{C-ED}$ and FlhA$_{C-EWD}$ were observed with an increase in the protein concentration (Fig. 7). However, these mutations caused fast-on and fast-off binding profiles (Fig. 7). Assuming that GST-FlgN binds to FlhA$_{C-ED}$ or FlhA$_{C-EWD}$ by inducing the dissociation of FlhA$_L$ with either of these $flhA$ mutations from the chaperone-binding site so that GST-FlgN forms a complex with FlhA$_{C-ED}$ or FlhA$_{C-EWD}$ on the sensor chip, their BLI data fitted well with a two-state reaction model, giving $K_D$ values of $37.8 ± 1.9 \, μM$ ($n = 3$) and $31.2 ± 1.1 \, μM$ ($n = 3$) for the FlgN–FlhA$_{C-ED}$ and FlgN–FlhA$_{C-EWD}$ interactions, respectively.

We next investigated whether deletion of FlhA$_L$ affect the binding process of FlgN to FlhA$_C$. The association and dissociation processes of FlhA$_{C-ΔL}$ were clearly observed at protein concentrations above 5 μM, and the BLI signals for the FlgN–FlhA$_{C-ΔL}$ interaction were much stronger at the same protein concentrations compared to the FlgN–FlhA$_{C-ED}$ and FlgN–FlhA$_{C-EWD}$ interactions (Fig. 7). Furthermore, the association and dissociation profiles of FlhA$_{C-ΔL}$ were different from those of FlhA$_{C-ED}$ and FlhA$_{C-EWD}$. Its BLI data did not fit the global one-state association-then-dissociation model, but fitted with a heterogeneous reaction model, showing a $K_D$ value of $3.1 ± 0.1 \, μM$ ($n = 3$). Thus, the binding affinity of FlhA$_{C-ΔL}$ for FlgN was higher than those of FlhA$_{C-ED}$ and FlhA$_{C-EWD}$. This suggests that FlhA$_L$ with either $flhA_{ED}$ or $flhA_{EWD}$ mutation inhibits the binding of FlgN to FlhA$_C$. Because the binding affinity of FlhA$_{C-ΔL}$ for FlgN was much lower than that of

wild-type FlhA$_C$, we suggest that FlhA$_L$ is required to keep FlhA$_C$ in the open form so that FlgN can efficiently and stably bind to the well-conserved hydrophobic dimple of FlhA$_C$.

## Discussion

The FlhA$_C$ ring serves as the docking platform for flagellar export chaperones in complex with their cognate substrates and facilitates the export of filament-type proteins to form the filament at the hook tip after completion of hook assembly[24–27]. The FlhA$_C$ ring also ensures the strict order of flagellar protein export, thereby allowing the huge and complex flagellar structure to be built efficiently on the cell surface[10,11,30,36]. An interaction of FlhA$_L$ with its neighboring FlhA$_C$ subunit in the nonamer ring is required for the initiation of filament-type protein export upon completion of hook assembly[10]. However, it remained unclear how the FlhA$_C$ ring mediates such hierarchical protein export during flagellar assembly.

In this study, we first performed genetic analyses of the $flhA_{EWD}$ mutant and found that this mutation reduces the protein transport activity of the fT3SS significantly (Fig. 2b, c). We also found that both the $flhA_{EWD}$ mutation and deletion of FlhA$_L$ reduce the binding affinity of FlhA$_C$ for FliJ (Fig. 3). Because the interaction between FliJ and FlhA$_L$ is required for activation of the fT3SS[32], we propose that Glu-351, Trp-354, and Asp-356 of FlhA$_L$ is required for stable interaction of FlhA$_L$ with FliJ to fully activate the fT3SS to facilitate flagellar protein export.

It has been reported that either $flhA(D456V)$, $flhA(F459A)$, or $flhA(T490M)$ mutation in the flagellar chaperone-binding site in FlhA$_C$ increases the levels of FlgE and FliK secretion by the $\Delta fliH$-$fliI$ $flhB(P28T)$ mutant[36], suggesting that this chaperone-binding site is also involved in the export of hook-type substrates. Here, we found that the $flhA_{EWD}$ mutation significantly increased the secretion level of FlgE by a $\Delta flgM$ mutant (Fig. 2b, c), thereby producing longer hooks (Fig. 4). This indicates that the $flhA_{EWD}$ mutation affects the termination of hook-type protein export, suggesting that an interaction between FlhA$_L$ and the chaperone-binding site of FlhA$_C$ coordinates the export of hook-type proteins with hook assembly in a highly organized and well-controlled manner. Furthermore, we also found that this triple mutation also reduced the secretion levels of filament-type substrates significantly (Fig. 2b, c), thereby reducing the number of flagellar filaments per cell (Supplementary Figs. 1 and 2). Taken all together, we propose that FlhA$_L$ serves as a structural switch for substrate specificity switching of the fT3SS from hook type to filament type and that Glu-351, Trp-354, and Asp-356 of FlhA$_L$ are directly involved in this export switching mechanism.

It has been reported that the $flhA_W$, $flhA_{ED}$, and $flhA_{EWD}$ mutations inhibit interactions between FlhA$_C$ and flagellar chaperones in complex with their cognate filament-type substrates[10], suggesting that FlhA$_L$ regulates the binding affinity of FlhA$_C$ for flagellar chaperones. The crystal structure of FlhA$_{C-ED}$ we solved in this study showed that FlhA$_L$ of a Mol-A molecule bound to the hydrophobic dimple of the flagellar chaperone-binding site of its nearest Mol-A in the crystal (Fig. 6). Although the relative orientations of these Mol-A molecules in the crystal differs from those in the FlhA$_C$ nonameric ring, FlhA$_L$ should be able to bind to the hydrophobic dimple of FlhA$_C$ in the nonamer ring structure as well because of a highly flexible nature of FlhA$_L$ (Fig. 8). The C-terminal region of FlhA$_L$ is flexible enough to allow such subunit orientations without changing the essential interaction between FlhA$_L$ and the chaperone-binding site of FlhA$_C$ (Fig. 8), as it has been shown to have various conformations in the known FlhAc structures[28]. BLI measurements indicated that FlhA$_L$ with either $flhA_{ED}$ or $flhA_{EWD}$ mutation inhibits the docking process of FlgN to FlhA$_C$ (Fig. 7). Because we also found that FlhA$_L$ is

required for stable interaction between FlgN and FlhA$_C$ (Fig. 7), we propose that the interaction between FlhA$_L$ and the hydrophobic dimple of its neighboring FlhA$_C$ subunit suppresses the docking of flagellar chaperones to the FlhA$_C$ ring platform during HBB assembly and that the hook assembly completion induces the detachment of FlhA$_L$ from the dimple through an interaction between FliK$_C$ and FlhB$_C$ and its attachment to the D1 and D3 domains to induce structural remodeling of the entire FlhA$_C$ ring, thereby terminating hook assembly and initiating filament formation (Fig. 8). Because FlhA$_{C-ED}$ monomer adopts a more compact conformation compared with the wild-type FlhA$_C$ monomer as judged by SEC (Fig. 5a), FlhA$_L$ may bind to FlhA$_C$ in a $cis$ manner as well. Therefore, it is also possible that FlhA$_L$ may block the docking of the flagellar chaperones to FlhA$_C$ by covering the binding site of the same FlhA$_C$ molecule.

## Methods

**Bacterial strains, plasmids, transductional crosses, and DNA manipulations**. Bacterial strains and plasmids used in this study are listed in Table 2. P22-mediated transductional crosses were performed with P22HT$int$. DNA manipulations were performed using standard protocols[38]. Site-directed mutagenesis were carried out using the QuikChange site-directed mutagenesis method as described in the manufacturer's instructions (Stratagene). DNA sequencing reactions were carried out using BigDye v3.1 (Applied Biosystems) and then the reaction mixtures were analyzed by a 3130 Genetic Analyzer (Applied Biosystems).

**Motility assays**. We transformed *Salmonella enterica* strains NH001 and NH001gM with a pTrc99A-based plasmid encoding wild-type FlhA or its mutant variant. Fresh transformants were inoculated into soft agar plates [1% (w/v) tryptone, 0.5% (w/v) NaCl, 0.35% Bacto agar] containing 100 µg ml$^{-1}$ ampicillin and incubated at 30 °C. At least five independent measurements were performed.

**Secretion assays**. *S. enterica* cells were grown in T-broth [1% (w/v) tryptone, 0.5% (w/v) NaCl] containing ampicillin at 30 °C with shaking until the cell density had reached an OD$_{600}$ of ca. 1.4–1.6. Cultures were centrifuged to obtain cell pellets and culture supernatants. The cell pellets were resuspended in a sample buffer solution [62.5 mM Tris-HCl, pH 6.8, 2% SDS, 10% glycerol, 0.001% bromophenol blue] containing 1 µl of 2-mercaptoethanol. Proteins in the culture supernatants were precipitated by 10% trichloroacetic acid and suspended in a Tris/SDS loading buffer (one volume of 1 M Tris, nine volumes of 1× sample buffer solution)[39] containing 1 µl of 2-mercaptoethanol. Both whole cellular proteins and culture supernatants were normalized to a cell density of each culture to give a constant cell number. After boiling proteins in both whole cellular and culture supernatant fractions at 95 °C for 3 min, these protein samples were separated by SDS-PAGE (normally 12.5% acrylamide) and transferred to nitrocellulose membranes (Cytiva) using a transblotting apparatus (Hoefer). Then, immunoblotting with polyclonal anti-FlgD, anti-FlgE, anti-FlgK, anti-FliC, anti-FliD, anti-FliI, or anti-FliK antibody was carried out using iBand Flex Western Device (Thermo Fisher Scientific). Detection was performed with Amersham ECL Prime western blotting detection reagent (Cytiva). Chemiluminescence signals were captured by a Luminoimage analyzer LAS-3000 (GE Healthcare). The band intensity of each blot was analyzed using an image analysis software, CS Analyzer 4 (ATTO, Tokyo, Japan). More than three independent experiments were performed.

**Electron microscopy observation of negatively stained *Salmonella* cells**. *S. enterica* cells were exponentially grown in 5 ml L-broth [1% (w/v) tryptone, 0.5% (w/v) yeast extract, 0.5% (w/v) NaCl] containing ampicillin at 30 °C. Five microliters of the cell culture was applied to carbon-coated copper grids and then negatively stained with 0.5% (w/v) phosphotungstic acid, pH 6.5. Micrographs were recorded at a magnification of ×1200 with a JEM-1010 transmission electron microscope (JEOL) operating at 100 kV.

**Observation of flagellar filaments with a fluorescent dye**. *S. enterica* cells were grown in T-broth containing ampicillin. The cells were attached to a coverslip (Matsunami glass, Japan), and unattached cells were washed away with motility buffer (10 mM potassium phosphate pH 7.0, 0.1 mM EDTA, 10 mM L-sodium lactate). Then, the flagellar filaments were labeled using anti-FliC antibody and anti-rabbit IgG conjugated with Alexa Fluor 594 (Invitrogen) as described previously[40]. After washing twice with the motility buffer, epi-fluorescence of Alexa Fluor 594 was observed by an inverted fluorescence microscope (IX-83, Olympus) with a ×150 oil immersion objective lens (UApo150XOTIRFM, NA 1.45, Olympus) and an Electron-Multiplying Charge-Coupled Device camera (iXon$^{EM}$ + 897-BI, Andor Technology)[41]. Fluorescence images were analyzed using ImageJ software version 1.52 (National Institutes of Health).

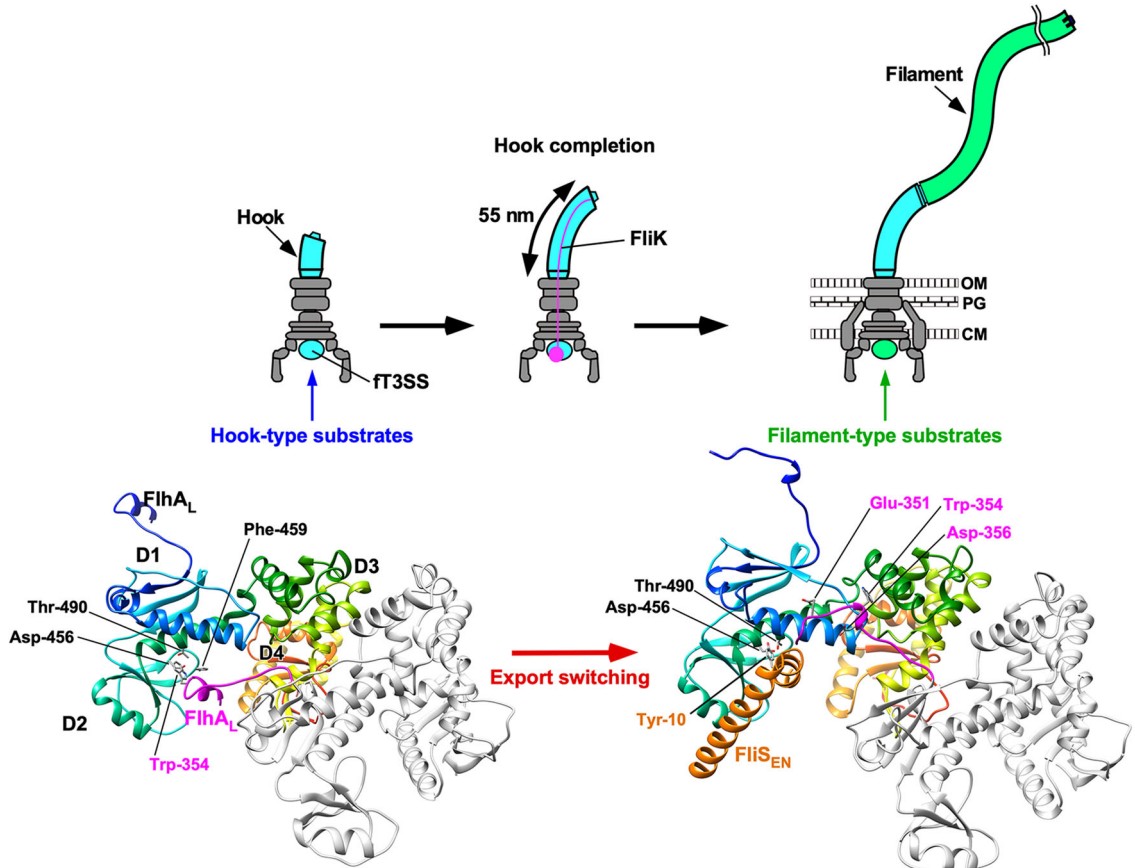

**Fig. 8 Structural rearrangements of FlhA_L responsible for export switching of fT3SS.** Trp-354 of FlhA_L binds to a well-conserved hydrophobic dimple containing Asp-456, Phe-459, and Thr-490 of its neighboring FlhA_C subunit in the FlhA_C ring not only to inhibit the interaction of FlhA_C with flagellar chaperones in complex with their cognate filament-type substrates but also to facilitate the export of the hook protein during hook assembly. When the hook reaches its mature length of about 55 nm, an interaction between FliK_C and FlhB_C triggers a conformational rearrangement of the FlhA_C ring so that FlhA_L dissociates from the hydrophobic dimple and binds to the D1 and D3 domains of the neighboring FlhA_C subunit, allowing the chaperones to bind to FlhA_C to facilitate the export of their cognate substrates for filament assembly.

## Table 2 Strains and plasmids used in this study.

| Strain/plasmid | Relevant characteristics | References |
|---|---|---|
| *E. coli* | | |
| BL21 Star (DE3) | Overexpression of proteins | Novagen |
| *Salmonella* | | |
| NH001 | Δ*flhA* | 47 |
| NH001gM | Δ*flhA* Δ*flgM::km* | This study |
| YI1003-xx | Pseudorevertants isolated from NH001 carrying pYI003 | This study |
| *Plasmids* | | |
| pTrc99AFF4 | Expression vector | 48 |
| pMM130 | pTrc99AFF4/FlhA | 49 |
| pMMGN101 | pGEX-6p-1/GST-FlgN | 25 |
| pMMJ1002 | pGEX-6p-1/GST-FliJ | 50 |
| pYI003 | pTrc99AFF4/FlhA(E351A/W354A/D356A) | 10 |
| pYI008 | pET15b/His-FlhA_C (residues 328–692 of FlhA) | 10 |
| pYI009 | pET15b/His-FlhA_C(W354A) | 10 |
| pYI010 | pET15b/His-FlhA_C(E351A/D356A) | 10 |
| pYI012 | pET15b/His-FlhA_C(E351A/W354A/D356A) | 10 |
| pYI008(F459C) | pET15b/His-FlhA_C(F459C) | 30 |
| pYI008(K548C) | pET15b/His-FlhA_C(K548C) | 30 |
| pYI008(F459C/K548C) | pET15b/His-FlhA_C(F459C/K548C) | 30 |
| pYI009(F459C/K548C) | pET15b/His-FlhA_C(W354A/F459C/K548C) | This study |
| pYI010(F459C/K548C) | pET15b/His-FlhA_C FlhA_C(E351A/D356A/F459C/K548C) | This study |
| pYI012(F459C/K548C) | pET15b/ His-FlhA_C(E351A/ W354A/D356A/F459C/K548C) | This study |
| pMKMhA008-1 | pET15b/His-FlhA_C lacking FlhA_L (residues 328–361) | This study |

**Bio-layer interferometry.** His-FlhA_C and its mutant variants were purified by Ni affinity chromatography, followed by SEC as described previously[28]. GST-FliJ and GST-FlgN were purified by GST affinity chromatography as described previously[25,28]. Purified protein samples were dialyzed overnight against a kinetic buffer [PBS (8.8 g of NaCl, 0.2 g of KCl, 3.63 g of $Na_2HPO_4 \cdot 12H_2O$, 0.24 g of $KH_2PO_4$, pH 7.4 per liter), 0.1% bovine serum albumin, 0.002% Tween-20] at 4 °C with three changes of PBS.

BLI measurements were carried out using a BLItz (FortéBio). GST-FliJ or GST-FlgN was immobilized to an anti-GST sensor tip (FortéBio). The sensor tip was then dipped into His-FlhA_C or its mutant variants to measure association before being dipped into the kinetic buffer to measure dissociation. Data were reference subtracted and fit to various model using BLItz Pro software (FortéBio) and BIAevaluation software (GE Healthcare).

**Hook length measurements.** The HBBs were purified from NH004gM carrying pMM130 or pYI003 as described previously[36]. *Salmonella* cells were grown in L-broth containing ampicillin at 30 °C with shaking until the cell density had reached an $OD_{600}$ of ca. 1.0. The cultures were centrifuged (10,000*g*, 10 min, 4 °C), and the cell pellets were suspended in 20 ml of ice-cold 0.1 M Tris-HCl pH 8.0, 0.5 M sucrose, followed by addition of EDTA and lysozyme at the final concentrations of 10 mM and 0.1 mg ml⁻¹, respectively. The cell suspensions were stirred for 30 min at 4 °C, and then were solubilized on ice for 1 h by adding Triton X-100 and MgSO₄ at final concentrations of 1% (w/v) and 10 mM, respectively. The cell lysates were adjusted to pH 10.5 with 5 M NaOH and then centrifuged (10,000*g*, 20 min, 4 °C) to remove cells debris. After ultracentrifugation (45,000*g*, 60 min, 4 °C), pellets were resuspended in 10 mM Tris-HCl, pH 8.0, 5 mM EDTA, 1% Triton X-100 and the solution was loaded a 20–50% (w/w) sucrose density gradient in 10 mM Tris-HCl, pH 8.0, 5 mM EDTA, 1% Triton X-100. After ultracentrifugation (49,100*g*, 13 h, 4 °C), intact flagella were collected and ultracentrifuged (60,000*g*, 60 min, 4 °C). Pellets were suspended in 50 mM glycine, pH 2.5, 0.1% Triton X-100 to depolymerize the flagellar filaments. After ultracentrifugation (60,000*g*, 60 min, 4 °C),

pellets were resuspended in 50 μl of 10 mM Tris-HCl, pH 8.0, 5 mM EDTA, 0.1% Triton X-100. The HBBs were negatively stained with 2% (w/v) uranyl acetate. Electron micrographs were recorded with a JEM-1011 transmission electron microscope (JEOL, Tokyo, Japan) operated at 100 kV and equipped with a F415 CCD camera (TVIPS, Gauting, Germany). Hook length was measured by ImageJ version 1.52 (National Institutes of Health).

**Size exclusion chromatography**. SEC was performed with a Superdex 75HR 10/30 column (GE Healthcare). Purified His-FlhA$_C$ and its mutant variants (10 μM) were run on the SEC column equilibrated with 50 mM Tri-HCl, pH 8.0, 150 mM NaCl at a flow rate of 0.5 ml min$^{-1}$. γ-Globulin (158 kDa), bovine serum albumin (66.4 kDa) and ovalbumin (43 kDa) were used as size markers. All fractions were run on SDS-PAGE and then analyzed by Coomassie Brilliant blue (CBB) staining.

**Native PAGE**. Purified His-FlhA$_C$ and its mutant variants (14.4 μM) were run on Native PAGE Novex Bis-Tris gels as described in the manufacturer's instructions (Invitrogen).

**Far-UV CD spectroscopy**. Far-UV CD spectroscopy of His-FlhA$_C$ or its mutant variants was carried out at room temperature using a Jasco-720 spectropolarimeter (JASCO International Co., Tokyo, Japan) as described previously[42]. The CD spectra of His-FlhA$_C$ and its mutant forms were measured in 20 mM Tris-HCl, pH 8.0 using a cylindrical fused quartz cell with a path length of 0.1 cm in a wavelength range of 200–260 nm. Spectra were obtained by averaging five successive accumulations with a wavelength step of 0.5 nm at a rate of 20 nm min$^{-1}$, response time of 8 s, and bandwidth of 2.0 nm.

**Cystein modification by mPEG-maleimide**. His-FlhA$_{C(F459C)}$, His-FlhA$_{C(K548C)}$, His-FlhA$_{C(F459C/K548C)}$, His-FlhA$_{C-W(F459C/K548C)}$, His-FlhA$_{C-ED(F459C/K548C)}$, and His-FlhA$_{C-EWD(F459C/K548C)}$ were dialyzed overnight against PBS (8 g of NaCl, 0.2 g of KCl, 3.63 g of Na$_2$HPO$_4$ 12H$_2$O, 0.24 g of KH$_2$PO$_4$, pH 7.4 per liter) at 4 °C. Twenty-five microliters of mPEG-maleimide reaction buffer (PBS containing 4 mM mPEG-maleimide) was added to 25 μl of 10 μM protein solutions. After incubation at 37 °C for 30 min, 5 μl of 2-mercaptoethanol was added to quench the reaction, and then 5 μl of 10% SDS was added. After centrifugation (20,000$g$, 20 min, 4 °C) to remove any aggregates, 60 μl of each soluble solution was mixed with 60 μl of 2× SDS loading buffer. After boiling at 95 °C for 3 min, each protein solution was run on SDS-PAGE and then analyzed by CBB staining.

**X-ray crystallographic study of FlhA$_C$(E351A/D356A)**. Initial crystallization screening was performed at 20 °C by the sitting-drop vapor-diffusion method using Wizard Classic I and II, Wizard Cryo I and II (Rigaku Reagents, Inc.), Crystal Screen, and Crystal Screen 2 (Hampton Research). Crystals suitable for X-ray analysis were obtained from drops prepared by mixing 0.5 μl protein solution with 0.5 μl reservoir solution containing 0.1 M Tris-HCl, pH 8.5, 20% (v/v) PEG 8000, and 200 mM MgCl$_2$. X-ray diffraction data were collected at synchrotron beamline BL41XU in SPring-8 (Harima, Japan) with the approval of the Japan Synchrotron Radiation Research Institute (JASRI) (Proposal No. 2016B2544 and 2018A2568). The FlhA$_C$(E351A/D356A) crystal was soaked in a solution containing 90% (v/v) of the reservoir solution and 10% (v/v) glycerol for a few seconds and was directly transferred into liquid nitrogen for freezing. The X-ray diffraction data were collected at the wavelength of 1.000 Å under nitrogen gas flow at 100 K. The diffraction data were processed with MOSFLM[43] and were scaled with Aimless[44]. The initial phase was determined by molecular replacement using the software package Phenix[45] with the wild-type FlhA$_C$ structure in the orthorhombic crystal form (PDB code: 6AI0) as a search model. The atomic model was constructed with Coot[46] and refined with Phenix[45]. During the refinement process, iterative manual modification was performed. The Ramachandran statistics indicated that 96.0%, 3.9%, and 0.1% residues were in the most favorable, allowed, and outlier regions, respectively. The diffraction data statistics and refinement statistics are summarized in Table 1.

**Statistics and reproducibility**. Statistical tests, sample size, and number of biological replicates are reported in the figure legends. Statistical analyses were done using KaleidaGraph software (HULINKS). Comparisons between datasets were performed using a two-tailed Student's $t$-test. A $P$ value of <0.05 was considered to be statistically significant difference. *$P < 0.05$; **$P < 0.01$; ***$P < 0.001$.

**Reporting summary**. Further information on research design is available in the Nature Research Reporting Summary linked to this article.

## Data availability
The X-ray crystal structure and structure factors of FlhA$_C$(E351A/D356A) have been deposited in Protein Data Bank under the accession code 7CTN. All data generated during this study are included in this published article, Supplementary Information and Supplementary Data file. Strains, plasmids, polyclonal antibodies, and all other data are available from the corresponding author on reasonable request.

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

## Acknowledgements
We thank beamline staffs at SPring-8 for technical help in use of beamlines BL41XU. This work was supported in part by JSPS KAKENHI Grant Numbers JP18K14638 and JP20K15749 (to M.K.), JP16J01859 (to N.T.), 25000013 (to K.N.), 15H02386 (to K.I.), and JP26293097 and JP19H03182 (to T.M.). This work has also been partially supported by JEOL YOKOGUSHI Research Alliance Laboratories of Osaka University to K.N.

## Author contributions
K.N., K.I., and T.M. conceived and designed research; Y.I., M. Kinoshita, M. Kida, N.T., K.I., and T.M. preformed research; Y.I., M. Kinoshita, M. Kida, N.T., K.I., and T.M. analyzed the data; and K.N., K.I., and T.M. wrote the paper based on discussion with other authors.

## Competing interests
The authors declare no competing interests.
