## [Peer Review File · Communications Biology]

Reviewers' comments:

Reviewer #1 (Remarks to the Author):

Inoue et al presented detailed analyses of the flexible linker of the major flagellar export apparatus protein FlhA (FlhAL) and its potential function in flagellar assembly. Given that FlhA is probably the most important protein in flagellar assembly and the FlhAL is the bridge between the cytoplasmic domain and the transmembrane domain and remains poorly understood, the studies presented in the manuscript will be valuable for the understanding of this important protein and the whole export apparatus. The major concerns are that the manuscript and the experimental results were poorly presented and written. In current format, it's very difficult to differentiate the results from the background information and the interpretation.

Specific comments

1. Figure 1 should be a supplemental information or as a part of Figure 2.
2. The experiments in Figure 2 are solid. However, the images used to measure the hook length should be included.
3. The claims of conformational change are not supported by any of the experiments. Additional experiments are needed if the authors want to make these claims.
4. Figure of crystal structure should be changed to clearly show what is being described.

Reviewer #2 (Remarks to the Author):

This paper attempts to show that a linker segment of FlhA interacts with a hydrophobic region of an adjacent FlhA subunit (a region known to be involved in the binding of chaperones for late-cargo export), and by doing so interferes with export of late cargoes. It is hypothesized that this interaction plays a key role in regulating the switch between early and late export cargoes. The proposal is based on mutant phenotypes, and on a crystal structure of a mutant variant of the protein. The idea is interesting and might account for the results, but overall, I find the case to be not very persuasive. Specific issues are detailed below.

Specific suggestions and concerns:

1. In the abstract, an important mutant is mentioned, and the mutational changes given. But what is the phenotype? Is this a locked-early mutant, for example?
2. line 49: isn't it the case that there are several more 'early' substrates? Why has it become conventional to call them 'hook type' when the early substrates also include the components of the rod?
3. line 80 introduces the mutant phenotypes, which is welcome, but I found the description incomplete: the hook length is not regulated properly—does this mean the hooks are too long, too short, or just highly variable?
4. line 81: In the mutants, the interaction with chaperones is inhibited. Here, the reader might like to know the nature of the evidence for this: How was binding (and its inhibition) measured? I.e., it would be helpful to briefly describe the observations at the foundation of this important statement.
5. line 86: 'do' should be 'does'
6. The pseudorevertants seem to more or less fully knock out FlgM. Levels of FlgK and FliD are measured to be higher, which is expected. Then, it is stated that the flgM mutations increase cytoplasmic levels of FliH, FliI, FliJ, and export substrates, but I don't see the data to support this. Is it just a proposal? The subsequent suggestion concerning the activation of the apparatus to be pmf driven seems to me a fairly large leap—i.e. I don't feel that it follows in a convincing way from what's actually been shown.
7. line 115. I found this phrasing a little confusing. The finding is that the pseudorevertants do not

fully rescue switching. As stated, it sounds like they cause the switching defect in the first place.

8. line 130. Why conformational rearrangements specifically? This didn't seem clear to me. Also, are there any known instances where a defect in switching interferes with filament assembly but does not also give rise to longer hooks? I.e. doesn't the process of switching specificity, so far as we know, do both things—shut down early export and initiate late export?

9. line 138. the triple, like the single and double mutations, reduces binding of chaperones. but the Trp residue, in the present model, is involved in blocking chaperone binding, right? So its loss might be expected to permit binding? Unless what happens in the triple mutant is that the relationship between adjacent FlhAc domains is altered, and this relationship is important for chaperone binding; i.e., binding is near the interface between subunits. line 139 introduces the idea of binding to the dimple, but it's not entirely clear how this follows from what's gone before.

10. If it's a conformational equilibrium in the mostly folded protein, then wouldn't two bands on SDS-PAGE (where the protein is largely unfolded) be unexpected? The idea seems to be that the relevant regions of the protein remain folded, and capable of the hypothesized interaction, even in SDS-PAGE. An alternative possibility is that a part of the protein binds SDS differently (more) in the mutants; this might involve a local partially folded conformation, but might not (and so might be more likely).

11. line 162. But the triple mutant still shows some of the compact conformation, even though the side chain of Trp 354 is changed to a methyl group. So the Trp is not needed.

12. The crystal structure shows an interaction in which a hydrophobic residue introduced by mutation (Ala 351) makes a hydrophobic contact. The alanine replaces a residue that is polar in the wild-type protein. Doesn't this argue that the interaction is an artifact of the hydrophobicity-increasing mutation, together with the crowded conditions in a crystal that practically force interactions to occur? It's difficult to see, from the views shown in fig. 4, what would be the effect of having residues 351 and 356 still being acidic (wild type), but it seems likely that they would interfere. It can also be noted that the introduced Ala residues will increase the helical propensity of this segment very appreciably. So even if the alanine at 356 doesn't interact directly at the hypothesized binding site, it could help stabilize the helical conformation that is involved in the binding seen in the crystal.

13. Looking at the structure, it seems that the spacing of the two domains is fairly large, and it emerges that while the interaction involving the linker occurs there, the other, better established interactions between adjacent subunits actually do not occur—i.e. the relative placement of these subunits is dictated by the crystal. This heightens the suspicion that the interaction involving the linker is not a native one.

Summary: My overall view is that the mutant phenotypes don't point convincingly toward the specific idea proposed here, and the interaction seen in the crystal structure could be an artifact of crystallization and the non-native hydrophobicity introduced in key positions. I feel that the study needs more direct evidence in support of the proposed interaction, in the flagellum rather than in a crystal. This would ideally include evidence that the interaction is lost upon switching from early to late export substrates.

Reviewer #3 (Remarks to the Author):

Inoue et al. present a paper titled "The FlhA linker mediates flagellar protein export switching during flagellar assembly" in which they present genetic, biochemical and structural data to support a role of the linker between the membrane embedded N-terminal and cytoplasmic C-terminal domains of FlhA in regulating the switch between secretion of hook-type to filament-type components.

Several FlhAL mutants impact flagellar filament assembly including E351A/D356A and W354A, which produce HBBs without filaments and hook length not controlled properly, and

E351A/W354A/D356A which does not produce HBBs at all. The authors initially identified some genetic pseudorevertants that restore some level of motility to a flhA(E351A/W354A/D356A) mutant. These commonly increased cellular levels of other components involved in secretion and it was interpreted that this bypasses the reduced affinity between FlhAc(E351A/W354A/D356A) and FliJ (a published observation), which is present at higher intracellular concentration. They go on to characterize several mutants - the triple mutant, E351A/D356A and W354A - with gel-filtration, gel-mobility shifts, methoxypolyethylene modification and an X-ray structure. The gel-filtration results show a small shift in elution volume for the mutants which is interpreted to suggest the linker mutants result in a more compact monomeric structure due to binding of the mutated linker to the hydrophobic dimple in a cis manner. This binding site is also shared with flagellar chaperones so linker binding effectively inhibits chaperone binding. A more compact nature is also proposed based on mobility in SDS-PAGE gels with the mutants showing faster migration. Ultimately, the determined structure of E351A/D356A does show the linker of one monomer in the crystal asymmetric unit bound to a neighbouring "hydrophobic dimple" similar to the interaction observed for flagellar chaperones; however the relative disposition the two molecules in the presented crystal is different to that in the monomeric ring. This is accounted for by the flexibility of the linker.

Overall, the manuscript is clearly written and presents an interesting structure capturing the FlhA linker bound to a neighbouring monomer, and model that offers a plausible explanation of the genetic data on hook length control. However, the accompanying biochemical and biophysical data does not convincingly support the conclusions drawn. The gel-filtration results and interpretation would benefit from further experiments to support the conclusions reached. For example, all samples should be run at the same concentration. FlhA homologues undergo transient self-association that can result in a concentration dependent change in elution profile with gel-filtration. Further, if the mutants affect self-association, which the linker is involved in, then this combination of different concentrations and mutants could account for the changes in elution position. Light scattering experiments (SEC-MALS and DLS) might provide a way to confirm the monomeric state and hydrodynamic properties of these constructs in solution. The different migration of the mutants in SDS-PAGE is surprising given these are denaturing gels. Running both boiled and unboiled samples might be further informative. Similarly the double band in the SDS-PAGE of the triple mutant (E351A/W354A/D356A), which doesn't appear to be present in the corresponding double Cys mutant used for the mPEG-maleimide modification, should be expected to disappear with boiling prior to SDS-PAGE if two different structural states. It is also somewhat confusing why the gel-filtration data is presented as suggesting an interaction of the linker with self, while the structure and model present an interaction with neighbouring molecule in the FlhA monomer, this is briefly referenced in the very last sentence. The manuscript would additionally benefit from some expanded discussion on the respective interactions of the chaperones/FliJ with FlhA, which contribute to the temporal and specific regulation of substrate secretion, especially in light of the recent structure the complex between Chlmydia SctV and SctO.

Given the broad interest and novelty of the structure, the study merits publication in Communications Biology providing the outlined issues with the supporting biochemical data are addressed.

Additional points

Table 1: Data looks like it extended further than presented eg CC1/2 in highest resolution bin is 0.915 and I/sig 2.8?

Figure 2: Isolation of pseudorevertants

mPEG modification - seems much more than 5kDa, is this just a change in gel migration?

Our responses are listed below. We highlighted all changes in red in the revised manuscript (Marked Up version).

To Reviewer #1

Inoue et al presented detailed analyses of the flexible linker of the major flagellar export apparatus protein FlhA (FlhAL) and its potential function in flagellar assembly. Given that FlhA is probably the most important protein in flagellar assembly and the FlhAL is the bridge between the cytoplasmic domain and the transmembrane domain and remains poorly understood, the studies presented in the manuscript will be valuable for the understanding of this important protein and the whole export apparatus. The major concerns are that the manuscript and the experimental results were poorly presented and written. In current format, it's very difficult to differentiate the results from the background information and the interpretation.

Re: Thank you so much for your comments. We agree that the background information for the experimental design may have been rather limited for readers to fully understand the purpose of this study. We added necessary background information in the Introduction and Results. We also tried to make the description of results clearly differentiated from the background information and interpretation. We hope the presentation of the results are sufficiently clearer in the revised manuscript.

Specific comments

1. Figure 1 should be a supplemental information or as a part of Figure 2.

Re: The original Fig. 1 has been added to new Fig. 2 of the revised manuscript.

2. The experiments in Figure 2 are solid. However, the images used to measure the hook length should be included.

Re: We added the EM images of HBBs to new Figure 4 of the revised manuscript.

3. The claims of conformational change are not supported by any of the experiments. Additional experiments are needed if the authors want to make these claims.

Re: We carried out Bio-Layer Interferometry (BLI) measurements to understand why the FlhA linker with either ED or EWD mutation inhibits the binding of the FlgN chaperone to the flagellar chaperone-binding site of FlhA_C (**Please see new Figure 7 of the revised manuscript**). We found that these mutations significantly affect the docking process of FlhA_C to immobilized GST-FlgN. We also found that complete deletion of the FlhA linker region (FlhA_L) increases the binding affinity of FlhA_C for FlgN significantly compared to FlhA_{C-ED} and FlhA_{C-EWD}. These results suggest that FlhA_L with these mutations inhibits the binding of FlgN to the chaperone-binding site of FlhA_C. Furthermore, we found that the binding affinity of the linker deletion mutant for FlgN was 30-fold lower than that of wild-type FlhA_C, suggesting that FlhA_L is required to keep FlhA_C in the open form so that FlgN can efficiently and stably bind to the chaperone-binding site of FlhA_C. Because FlhA_L binds to its nearest FlhA_C subunit to promote the initiation of filament-type protein export (**Terahara et al. Sci. Adv. 2018; Inoue et al. Structure 2019**), we would like to keep our present model shown in new Figure 8 of the revised manuscript.

4. Figure of crystal structure should be changed to clearly show what is being described.

Re: We modified it to make non-expert readers readable.

To Reviewer #2

This paper attempts to show that a linker segment of FlhA interacts with a hydrophobic region of an adjacent FlhA subunit (a region known to be involved in the binding of chaperones for late-cargo export), and by doing so interferes with export of late cargoes. It is hypothesized that this interaction plays a key role in regulating the switch between early and late export cargoes. The proposal is based on mutant phenotypes, and on a crystal structure of a mutant variant of the protein. The idea is interesting and might account for the results, but overall, I find the case to be not very persuasive. Specific issues are detailed below.

Re: Thank you so much for all your comments and suggestions to improve our manuscript.

Specific suggestions and concerns:

1. *In the abstract, an important mutant is mentioned, and the mutational changes given. But what is the phenotype? Is this a locked-early mutant, for example?*

Re: Agreed and mentioned the phenotype of our *flhA* linker mutant in the Abstract as follows:

“To address how Flh_L brings the order to flagellar assembly, we analyzed the *flhA(E351A/W354A/D356A) ΔflgM* mutant and found that this triple mutation in Flh_L increased the secretion level of hook protein by 5-fold, thereby increasing hook length.”

2. *line 49: isn't it the case that there are several more 'early' substrates? Why has it become conventional to call them 'hook type' when the early substrates also include the components of the rod?*

Re: Thank you for the point. You are correct that the flagellar building blocks are classified into two classes: one is the rod-type (FliE, FlgB, FlgC, FlgF, FlgG, FlgJ) and hook-type class (FlgD, FlgE and FliK) needed for the assembly of the rod and hook, and the other is the filament-type class (FlgK, FlgL, FlgM, FliC and FliD) responsible for filament assembly at the hook tip. We mentioned this point in the Introduction.

3. *line 80 introduces the mutant phenotypes, which is welcome, but I found the description incomplete: the hook length is not regulated properly—does this mean the hooks are too long, too short, or just highly variable?*

Re: We modified the sentence as follows:

“The *flhA(E351A/D356A)* (hereafter referred to as *flhA_{ED}*) and *flhA(W354A)* (hereafter referred to as *flhA_W*) mutants produces the HBBs without the filament attached¹⁰. Hook lengths of the *flhA_{ED}* and *flhA_W* mutants are 54.0 ± 22.3 nm [mean \pm standard deviation (SD)] and 52.9 ± 19.9 nm, respectively, where their SD values are larger than the wild-type one (51.0 ± 6.9 nm), indicating their hook length is not controlled properly¹⁰.”

4. *line 81: In the mutants, the interaction with chaperones is inhibited. Here, the reader*

might like to know the nature of the evidence for this: How was binding (and its inhibition) measured? I.e., it would be helpful to briefly describe the observations at the foundation of this important statement.

Re: We modified the sentence as follows:

“Pull-down assays by GST affinity chromatography have revealed that the *flhA_{ED}* and *flhA_W* mutations reduce the binding affinity of FlhAc for flagellar chaperones in complex with their cognate filament-type substrates¹⁰.”

5. *line 86: ‘do’ should be ‘does’*

Re: Corrected

6. *The psuedorevertants seem to more or less fully knock out FlgM. Levels of FlgK and FliD are measured to be higher, which is expected. Then, it is stated that the flgM mutations increase cytoplasmic levels of FliH, FliI, FliJ, and export substrates, but I don’t see the data to support this. Is it just a proposal? The subsequent suggestion concerning the activation of the apparatus to be pmf driven seems to me a fairly large leap—i.e. I don’t feel that it follows in a convincing way from what’s actually been shown.*

Re: We showed that FlgM deletion increased the cellular levels of flagellar structural subunits and the FliI ATPase (**Please see new Figure 2b of the revised manuscript**). We also modified our description to make our proposal clear as follows:

“The interaction of FliJ with FlhAL is required for activation of the fT3SS, and FliH and FliI are required for efficient interaction between FliJ and FlhAL³². It has been reported that over-expression of export substrates and FliJ by FlgM deletion overcomes the loss of both FliH and FliI to a considerable degree³³. Because the *flhA_{EWD}* mutation reduces the binding affinity of FlhAc for FliJ¹⁰, this suggests that these *flgM* mutations increase the cytoplasmic levels of FliH, FliI, FliJ and export substrates to allow the *flhA_{EWD}* mutant to export flagellar building blocks for producing a small number of flagella on the cell surface. Therefore, we propose that Glu-351, Trp-354 and Asp-356 of FlhAL also play an important role in the activation mechanism of the fT3SS.”

7. line 115. I found this phrasing a little confusing. The finding is that the pseudorevertants do not fully rescue switching. As stated, it sounds like they cause the switching defect in the first place.

Re: We carried out quantitative immunoblotting to quantify the secretion levels of flagellar structural proteins by the $\Delta flgM$ and $flhA_{EWD} \Delta flgM$ strains and found that the secretion level of FlgE by the $flhA_{EWD} \Delta flgM$ cells is about 5-fold higher than that by the $\Delta flgM$ cells (**Please see new Figure 2c of the revised manuscript**). Therefore, we modified our description as follows:

“We found that the $flhA_{EWD} \Delta flgM$ strain secreted a much larger amount of FlgE into the culture media than the $\Delta flgM$ strain, raising the possibility that the length of the hook produced by this mutant may be longer than the wild-type length.”

8. line 130. Why conformational rearrangements specifically? This didn't seem clear to me. Also, are there any known instances where a defect in switching interferes with filament assembly but does not also give rise to longer hooks? I.e. doesn't the process of switching specificity, so far as we know, do both things—shut down early export and initiate late export?

Re: An interaction between FliK and FlhB triggers export switching of the σ^{54} to shut down early export and initiate late export, thereby terminating hook assembly and initiating filament formation at the hook tip. So, many $fliK$ mutants and certain $flhB$ point mutants produce polyhooks without the filament attached. We have shown that hook lengths of the $flhA_{ED}$ and $flhA_W$ mutants are 54.0 ± 22.3 nm (mean \pm SD) and 52.9 ± 19.9 nm, respectively, where their SD values are larger than the wild-type one (51.0 ± 6.9 nm). Thus, their hook length is nearly the wild-type one although not controlled properly (**Terahara et al. Sci. Adv. 2018**). However, these $flhA$ linker mutants cannot produce the filaments (**Terahara et al. Sci. Adv. 2018**), suggesting that these two $flhA$ mutations inhibit the initiation of late export. Pull-down assays by GST affinity chromatography have shown that these mutations considerably reduce the binding affinity of FlhAc for the FlgN chaperone in complex with their cognate substrate FlgK (**Terahara et al. Sci. Adv. 2018**). Furthermore, high-speed atomic force microscopy has revealed that these linker mutations inhibit cooperative FlhAc ring formation on

mica surface (**Terahara et al. Sci. Adv. 2018**). These observations suggest that conformational rearrangements of FlhAc is required for the initiation of late export. To make it clearer, we modified our proposal as follows:

“Because high-speed atomic force microscopy has shown that the *flhA_{EWD}* mutation also inhibits highly cooperative FlhAc ring formation¹⁰, we propose that FlhA_L regulates the conformational rearrangement of FlhAc, which is required for efficient termination of hook assembly and efficient initiation of filament formation at the hook tip.”

9. line 138. the triple, like the single and double mutations, reduces binding of chaperones. but the Trp residue, in the present model, is involved in blocking chaperone binding, right? So its loss might be expected to permit binding? Unless what happens in the triple mutant is that the relationship between adjacent FlhAc domains is altered, and this relationship is important for chaperone binding; i.e., binding is near the interface between subunits. line 139 introduces the idea of binding to the dimple, but it's not entirely clear how this follows from what's gone before.

Re: In the crystal structure of FlhAc we previously reported (**Saijo-Hamano et al. Mol. Microbiol. 2010**), FlhAc is in the open form, and Trp-354 of FlhA_L binds to a hydrophobic pocket formed by Val-357 of FlhA_L and Phe-400 and Leu-401 in domain D1, Leu-512 in domain D2 and the side chain arms of Glu-508, Gln-511 and Arg-515 in domain D3 of its neighboring subunit, possibly representing the FlhAc ring conformation in the state of the filament-type substrate export. Because the W354A mutation significantly reduces the binding affinity of FlhAc for the FlgN chaperone in complex with FlgK and also inhibits highly cooperative FlhAc ring formation on mica surface (**Terahara et al. Sci. Adv. 2018**), we think that the interaction of Trp-354 with the hydrophobic pocket described above is critical for making the chaperone-binding site open to allow the flagellar chaperone to bind to FlhAc to facilitate the export of filament-type proteins. Although the EWD triple mutation seems to weaken the interaction between FlhA_L and the hydrophobic dimple of FlhAc, which we visualized in the present crystal, we think that the EWD mutation suppresses the conformational change of FlhAc necessary for the substrate specificity switching from the hook-type state to the filament-type one so that the flagellar chaperones cannot bind to the hydrophobic dimple efficiently (**Please see new Figure 8 of the revised manuscript**).

10. *If it's a conformational equilibrium in the mostly folded protein, then wouldn't two bands on SDS-PAGE (where the protein is largely unfolded) be unexpected? The idea seems to be that the relevant regions of the protein remain folded, and capable of the hypothesized interaction, even in SDS-PAGE. An alternative possibility is that a part of the protein binds SDS differently (more) in the mutants; this might involve a local partially folded conformation, but might not (and so might be more likely).*

Re: Although we really do not know why Flh_{AC-ED} showed a slightly faster mobility in SDS-PAGE gels and why Flh_{AC-EWD} showed two different bands on SDS-PAGE gels, with the slower mobility band corresponding to wild-type Flh_{AC} and the faster one corresponding to Flh_{AC-ED}, we assume that a strong hydrophobic interaction between Flh_{AL} with the ED mutation and the hydrophobic dimple may allow this contact region to partially remain folded even in the SDS-PAGE and that the EWD triple mutation presumably weakens this strong hydrophobic interaction between Flh_{AL} and the dimple, thereby causing two different bands on SDS-PAGE gels. We also agree with your alternative possibility.

11. *line 162. But the triple mutant still shows some of the compact conformation, even though the side chain of Trp 354 is changed to a methyl group. So the Trp is not needed.*

Re: Agreed and deleted this sentence.

12. *The crystal structure shows an interaction in which a hydrophobic residue introduced by mutation (Ala 351) makes a hydrophobic contact. The alanine replaces a residue that is polar in the wild-type protein. Doesn't this argue that the interaction is an artifact of the hydrophobicity-increasing mutation, together with the crowded conditions in a crystal that practically force interactions to occur? It's difficult to see, from the views shown in fig. 4, what would be the effect of having residues 351 and 356 still being acidic (wild type), but it seems likely that they would interfere. It can also be noted that the introduced Ala residues will increase the helical propensity of this segment very appreciably. So even if the alanine at 356 doesn't interact directly at the hypothesized binding site, it could help stabilize the helical conformation that is involved in the binding seen in the crystal.*

Re: We agree with this reviewer that the introduced Ala residues increase the helical propensity of residues 349–357 of Flh_{AL} and could help stabilize its helical conformation involved in the binding in the crystal. But the binding of a tryptophane residue to a hydrophobic pocket is strong enough to be typically used for specific binding of many protein partners. Also, when Ala-351 and Ala-356 of Flh_{AC-ED} in the crystal structure are replaced by original Glu-351 and Asp-356 residues, respectively, the side chain arm of Glu-351 makes a hydrophobic contact with Pro-442 (**Please see new Supplementary Fig. 6 of the revised manuscript**), suggesting that Flh_{AL} binding to the hydrophobic dimple of Flh_{AC} by Trp-354 is further stabilized by Glu-351. Therefore, we changed our description as follows:

“When Ala-351 and Ala-356 of Flh_{AC-ED} in the crystal structure were replaced back to the original Glu-351 and Asp-356 residues, respectively, the side chain arm of Glu-351 can form a hydrophobic contact with Pro-442 (Supplementary Fig. 6), suggesting that Flh_{AL} can bind to the hydrophobic dimple of Flh_{AC} even in the wild-type without the *flhA_{ED}* mutation. Because the introduced Ala residues would increase the helical propensity of residues 349–357 of Flh_{AL} as seen in the crystal, we suggest that the *flhA_{ED}* mutation allowed residues 349–357 of Flh_{AL} to efficiently form an α -helix to stabilize the binding of Flh_{AL} to the hydrophobic dimple of Flh_{AC}.”

13. Looking at the structure, it seems that the spacing of the two domains is fairly large, and it emerges that while the interaction involving the linker occurs there, the other, better established interactions between adjacent subunits actually do not occur—i.e. the relative placement of these subunits is dictated by the crystal. This heightens the suspicion that the interaction involving the linker is not a native one.

Re: To investigate whether the interaction of Flh_{AL} with the chaperone-binding site reflects the functional state of Flh_{AC}, we analyzed the interaction between Flh_{AC} and FlgN by BLI measurements. We found that the linker mutations inhibit the docking of Flh_{AC} to immobilized GST-FlgN. Deletion of Flh_{AL} increased the binding affinity of Flh_{AC} for FlgN compared to Flh_{AC} with either ED or EWD mutation. These results suggest that these Flh_{AL} linker mutations inhibit the binding of FlgN to the chaperone-binding site of Flh_{AC}. We also found that the *flhA_{EWD}* mutation significantly increases the secretion level of FlgE by a Δ *flgM* mutant by about 5-fold (**Please see Figure 2c of the revised manuscript**), thereby producing longer hooks (**Please see Fig. 4 of**

the revised manuscript). As mentioned above, we previously reported that the *flhA(D456V)*, *flhA(F459A)* and *flhA(T490M)* mutations in the flagellar chaperone-binding site in FlhAc increases the levels of FlgE and FliK secretion by the Δ *fliH-fliI flhB(P28T)* mutant strain by about 10-fold and 3-fold, respectively (**Inoue et al., Sci. Rep. 2018**), suggesting that this chaperone-binding site is also involved in the export of hook-type substrates. These genetic and biochemical data strongly suggest that the interaction between FlhAL and the chaperone-binding site of FlhAc coordinates the export of hook-type proteins with hook assembly in a highly organized and well-controlled manner. Therefore, we think that the interaction between FlhAL and the chaperone-binding site of FlhAc we observed in the crystal structure would reflect a functional state of the FlhAc ring during HBB assembly rather than an artifact of crystallization.

Summary: My overall view is that the mutant phenotypes don't point convincingly toward the specific idea proposed here, and the interaction seen in the crystal structure could be an artifact of crystallization and the non-native hydrophobicity introduced in key positions. I feel that the study needs more direct evidence in support of the proposed interaction, in the flagellum rather than in a crystal. This would ideally include evidence that the interaction is lost upon switching from early to late export substrates.

Re: As mentioned just above, all our additional data strongly suggest that the interaction of FlhAL with the chaperone-binding site reflects the functional state of FlhAc during HBB assembly. Glu-351, Trp-354 and Asp-356 of FlhAL also bind to the D1 and D3 domains of its neighboring FlhAc subunit to stabilize FlhAc ring structure, as shown in the crystal structure we reported previously (**Saijo-Hamano et al. Mol. Microbiol. 2010**), and such interactions allow flagellar chaperones to bind to FlhAc in the ring to facilitate filament-type export (**Terahara et al. Sci. Adv. 2018**).

To Reviewer #3

Inoue et al. present a paper titled "The FlhA linker mediates flagellar protein export switching during flagellar assembly" in which they present genetic, biochemical and structural data to support a role of the linker between the membrane embedded N-terminal and cytoplasmic C-terminal domains of FlhA in regulating the switch between secretion of hook-type to filament-type components.

Several FlhAL mutants impact flagellar filament assembly including E351A/D356A and W354A, which produce HBBs without filaments and hook length not controlled properly, and E351A/W354A/D356A which does not produce HBBs at all. The authors initially identified some genetic pseudorevertants that restore some level of motility to a flhA(E351A/W354A/D356A) mutant. These commonly increased cellular levels of other components involved in secretion and it was interpreted that this bypasses the reduced affinity between FlhAc(E351A/W354A/D356A) and FliJ (a published observation), which is present at higher intracellular concentration. They go on to characterize several mutants - the triple mutant, E351A/D356A and W354A - with gel-filtration, gel-mobility shifts, methoxypolyethylene modification and an X-ray structure. The gel-filtration results show a small shift in elution volume for the mutants which is interpreted to suggest the linker mutants result in a more compact monomeric structure due to binding of the mutated linker to the hydrophobic dimple in a cis manner. This binding site is also shared with flagellar chaperones so linker binding effectively inhibits chaperone binding. A more compact nature is also proposed based on mobility in SDS-PAGE gels with the mutants showing faster migration. Ultimately, the determined structure of E351A/D356A does show the linker of one monomer in the crystal asymmetric unit bound to a neighbouring "hydrophobic dimple" similar to the interaction observed for flagellar chaperones; however the relative disposition the two molecules in the presented crystal is different to that in the monomeric ring. This is accounted for by the flexibility of the linker.

Overall, the manuscript is clearly written and presents an interesting structure capturing the FlhA linker bound to a neighbouring monomer, and model that offers a plausible explanation of the genetic data on hook length control.

Re: Thank you so much for your supportive comments

However, the accompanying biochemical and biophysical data does not convincingly support the conclusions drawn. The gel-filtration results and interpretation would benefit from further experiments to support the conclusions reached. For example, all samples should be run at the same concentration. FlhA homologues undergo transient self-association that can result in a concentration dependent change in elution profile with gel-filtration. Further, if the mutants affect self-association, which the linker is

involved in, then this combination of different concentrations and mutants could account for the changes in elution position. Light scattering experiments (SEC-MALS and DLS) might provide a way to confirm the monomeric state and hydrodynamic properties of these constructs in solution.

Re: Thank you so much for your comments. Our SEC-MALS is quite old and unfortunately did not work at all. So, we could not carry out SEC-MALS. Xing *et al.* showed that FlhAc forms dimer in a protein concentration dependent manner and that FlhAL is required for its dimerization (**Xing *et al.*, Nat. Commun. 2018**). They showed that FlhAc elutes as dimer at a protein concentration of 0.2 mM but as monomer at a protein concentration of 0.05 mM under a physiological salt concentration of 100 mM. They also showed that FlhAc without FlhAL elutes as monomer even at a high protein concentration. Therefore, we carried out analytical size exclusion chromatography again at the same protein concentration of 10 μ M in the presence of 150 mM NaCl. We also used BSA (66.4 kDa) as a marker. New SEC data clearly showed that they elute as monomer (**Please see new Fig. 5a of the revised manuscript**).

The different migration of the mutants in SDS-PAGE is surprising given these are denaturing gels. Running both boiled and unboiled samples might be further informative. Similarly the double band in the SDS-PAGE of the triple mutant (E351A/W354A/D356A), which doesn't appear to be present in the corresponding double Cys mutant used for the mPEG-maleimide modification, should be expected to disappear with boiling prior to SDS-PAGE if two different structural states.

Re: We carried out native PAGE and found that FlhAc-ED showed a slightly faster mobility even in native gels (**Please see New Figure 5c of the revised manuscript**). Because the F459C substitution is located in the chaperone-binding site of FlhAc, we assume that the F459C substitution may stabilize a compact conformation of FlhAc-EWD so that the upper band is gone.

It is also somewhat confusing why the gel-filtration data is presented as suggesting an interaction of the linker with self, while the structure and model present an interaction with neighbouring molecule in the FlhA nonomer, this is briefly referenced in the very last sentence.

Re: We modified the last sentence as follows to make it clearer:

“Because FlhAc_{ED} monomer adopts a more compact conformation compared with the wild-type FlhAc monomer as judged by SEC (Fig. 5), FlhAL may bind to FlhAc in a cis manner as well. Therefore, it is also possible that FlhAL may block the docking of the flagellar chaperones to FlhAc by covering the binding site of the same FlhAc molecule.”

The manuscript would additionally benefit from some expanded discussion on the respective interactions of the chaperones/FliJ with FlhA, which contribute to the temporal and specific regulation of substrate secretion, especially in light of the recent structure the complex between Chlamydia SctV and SctO.

Re: Thank you so much for your comments. We analyzed the interaction of FlhAc with FliJ by BLI measurements and confirmed that FlhAL is required for the interaction with FliJ. We added new data to the revised manuscript to discuss the FliJ-FlhAc interaction as follows:

“Effect of deletion of FlhAL on the interaction between FlhAc and FliJ. The crystal structure of a FliJ homologue, CdsO, in complex with CdsV_C, which is a FlhAc homologue, has shown that CdsO binds to a large cleft between domains D4 of neighboring CdsV_C subunits in the CdsV_C ring structure but not to the linker region of CdsV_C³⁴. To confirm the importance of FlhAL in the interaction between FlhAc and FliJ, we analyzed the binding of FlhAc to immobilized GST-FliJ by Bio-Layer Interferometry (BLI) measurements³⁵. The FliJ-FlhAc interaction showed a complex binding profile (Fig. 3a) and did not fit the global one-state association-then-dissociation model. Assuming that FlhAc binds to GST-FliJ to form a GST-FliJ/FlhAc complex, followed by a conformational change of this complex, the BLI data fitted well with a two-state reaction model and provided a K_D value of about 1.3 μM (Fig. 3b). Unlike wild-type FlhAc, the association and dissociation processes of FlhAc with the *flhA_{EWD}* mutation (FlhAc_{EWD}) or FlhAc lacking FlhAL (FlhAc_{ΔL}) were observed at protein concentrations above 10 μM (Fig. 3a). Their association and dissociation processes were also different from those of wild-type FlhAc. The association profile of these mutant proteins was composed of two distinct (fast-on and slow-on) processes, and the dissociation profile was also composed of two distinct (fast-off and slow-off) processes. It has been shown that wild-type FlhAc forms dimer in a protein concentration dependent manner

and that FlhA_L is required for efficient dimerization of FlhA_C²⁷. So, their BLI data were fitted well with curves predicted by the Hill equation, with K_D values of about 60 μM and 49 μM for the FliJ-FlhA_{C-EWD} and FliJ-FlhA_{C-ΔL} interactions, respectively (Fig. 3b). Thus, both *flhA_{EWD}* mutation and deletion of FlhA_L significantly reduced the binding affinity of FlhA_C for FliJ. Therefore, we conclude that FlhA_L is required for stable interaction between FliJ and FlhA_C.”

Given the broad interest and novelty of the structure, the study merits publication in Communications Biology providing the outlined issues with the supporting biochemical data are addressed.

Re: Thank you so much for all your supportive and constructive comments.

Additional points

Table 1: Data looks like it extended further than presented eg CC1/2 in highest resolution bin is 0.915 and I/sig 2.8?

Re: We analyzed our X-ray data at higher resolution than 2.8 Å, but CC(1/2) and I/σI deteriorated and so we determined the FlhA_{C-ED} structure at 2.8 Å

Figure 2: Isolation of pseudorevertants

Re: Corrected (**Supplementary Fig. 1 of the revised manuscript**).

mPEG modification - seems much more than 5kDa, is this just a change in gel migration?

Re: We think so.

REVIEWERS' COMMENTS:

Reviewer #1 (Remarks to the Author):

My previous concerns have been addressed in the revised manuscript. It is now greatly improved to allow a better understanding of the largest membrane protein in the flagellar export apparatus.

Reviewer #3 (Remarks to the Author):

The authors have addressed adequately the prior concerns in this revision.

Our responses are listed below.

To Reviewer #1

My previous concerns have been addressed in the revised manuscript. It is now greatly improved to allow a better understanding of the largest membrane protein in the flagellar export apparatus.

Re: Thank you so much for all your helpful comments and suggestions for improving our manuscript.

To Reviewer #3

The authors have addressed adequately the prior concerns in this revision.

Re: Thank you very much for all your helpful comments and suggestions for improving our manuscript.